



# Pollution affects Arabian and Saharan dust optical properties in the Eastern Mediterranean

Marilena Teri[1,2], Josef Gasteiger[1,a], Katharina Heimerl[1,b], Maximilian Dollner[1], Manuel Schöberl[1,2], Petra Seibert[3,4], Anne Tipka[3,c], Thomas Müller[5], Sudharaj Aryasree[6], Konrad Kandler[6], and Bernadett Weinzierl[1]

[1]University of Vienna, Faculty of Physics, Aerosol Physics and Environmental Physics, 1090 Vienna, Austria
[2]University of Vienna, Vienna Doctoral School in Physics, 1090 Vienna, Austria
[3]University of Vienna, Department of Meteorology and Geophysics, 1090 Vienna, Austria
[4]University of Natural Resources and Life Sciences, Institute of Meteorology and Climatology, 1180 Vienna, Austria
[5]Leibniz-Institute for Tropospheric Research, Tropospheric Aerosols, 04318 Leipzig, Germany
[6]Institute of Applied Geosciences, Technical University Darmstadt, 64287 Darmstadt, Germany
[a]now at: Hamtec Consulting GmbH @ EUMETSAT, Darmstadt, Germany
[b]now at: Vrije Universiteit Amsterdam, Department of Earth Sciences, 1081 HV Amsterdam, The Netherlands
[c]now at: International Data Centre, Comprehensive Nuclear-Test-Ban Treaty Organization, PO Box 1200, 1400 Vienna, Austria

**Correspondence:** Bernadett Weinzierl (bernadett.weinzierl@univie.ac.at)

**Abstract.** Uncertainties in the mineral dust's direct radiative effect arise from the variability in its optical properties. The optical properties can also be influenced by mixing processes with anthropogenic aerosols, such as black carbon or fine particles (called "pollution" in this study). We aimed to investigate the effect of mixing pollution with mineral dust aerosols from different source regions on the intensive aerosol optical properties. Thus, the Ångström exponents of scattering and absorption
(i.e., their wavelength dependence), the single scattering albedo, and the asymmetry parameter were determined from direct optical measurements performed during the A-LIFE aircraft field experiment over the Eastern Mediterranean. This location provided access to Arabian and Saharan dust layers mixed with pollution. Our findings indicated significant changes in all the intensive aerosol optical properties with increasing pollution content within mineral dust layers. Interestingly, the differences between Arabian and Saharan dust's intensive aerosol optical properties were negligible. We discussed the implications of
these results for identifying mineral dust events and for their direct radiative effect. First, the mixing with pollution masked the mineral dust signal, suggesting that caution is needed when using the Ångström exponents for identifying mineral dust events. However, the Ångström exponents can help estimate the amount of pollution once a mineral dust event is confirmed. Second, our measurements of the asymmetry parameter and single scattering albedo changed from pure to polluted mineral dust layers (e.g., at $525\,\mathrm{nm}$, the median values decreased from $0.67$ to $0.56$ and from $0.96$ to $0.89$, respectively). These changes have
opposing effects on the short-wave direct radiative effect efficiency (i.e., the direct radiative effect per unit of aerosol optical depth) and may partly cancel out each other. Nevertheless, the impact of mixing with pollution on the mineral dust's direct radiative effect efficiency can differ depending on the surface albedo. In conclusion, accurate quantification of the pollution content within mineral dust layers is crucial. The pollution significantly impacts mineral dust event identification, its optical properties, and the local direct radiative effect.



## 1 Introduction

Large amounts of mineral dust particles are emitted into the atmosphere by winds blowing over arid regions worldwide (Choobari et al., 2014). Once in the atmosphere, they can form layers at high altitudes and travel long distances (Weinzierl et al., 2017; Adebiyi et al., 2023). Mineral dust particles interact directly with solar and terrestrial radiation via scattering and absorption, as well as indirectly by altering the cloud properties. Consequently, mineral dust particles affect the Earth's energy
budget and its climate system. The direct radiative effect of mineral dust is still uncertain due to significant uncertainties in its microphysical and optical properties (Kok et al., 2017; Li et al., 2021). This uncertainty contributes significantly to the overall uncertainty of the effective radiative effect of mineral dust (Kok et al., 2023).

The properties of mineral dust layers may be modified through aging and mixing processes (Riemer et al., 2019). For example, the mineral dust size distribution can change (e.g., Weinzierl et al., 2011) or the mineral dust layer can mix (e.g.,
Petzold et al., 2011; Denjean et al., 2016) with anthropogenic aerosol such as black carbon or fine particles (which we will refer to as "pollution" in this study). Especially, black carbon particles, emitted from fossil and biomass fuel combustion, exhibit a higher absorption efficiency and different spectral behavior than mineral dust particles (Bond et al., 2013; Caponi et al., 2017). Hence, black carbon particles within mineral dust layers can play a critical role in enhancing the mineral dust absorption properties and resulting in a relatively greater impact on atmospheric heating compared to low altitudes black carbon
particles, since their residence time at higher altitudes is longer (Seinfeld et al., 2004).

The aerosol direct radiative effect depends on the aerosol optical properties, in particular the particle extinction coefficient $\sigma_{ep}$ (the sum of the particle scattering and absorption coefficients), the asymmetry parameter $g$ (i.e., the cosine-weighted average of the angular distribution of the scattered light, Andrews et al., 2006), and the single scattering albedo SSA (the fraction of the extinct light that is scattered). These aerosol optical properties are generally derived from remote sensing
retrievals (e.g., Granados-Muñoz et al., 2019) or from in-situ measurements of the particle size distribution via optical modeling (e.g., Kok et al., 2017). These methods require assumptions about aerosol types, refractive indices, and mixing states.

Direct in-situ measurements of the particle scattering and absorption coefficients in the visible spectrum can be used to calculate intensive aerosol optical properties that allow a better understanding of the optical signature and the direct radiative effect of aerosol mixtures. These intensive aerosol optical properties include the wavelength dependence of the particle scattering
and absorption coefficients and of the single scattering albedo, known as the scattering, absorption and single scattering albedo Ångström exponents (SAE, AAE, and SSAAE), which provide near-real-time information on the aerosol types that impact the aerosol optical properties (e.g., Costabile et al., 2013; Cazorla et al., 2013; Cappa et al., 2016). Additionally, the asymmetry parameter ($g$) and the single scattering albedo (SSA) are crucial for determining the intrinsic short-wave direct radiative effect of the aerosol mixture (e.g., Sherman et al., 2015).

These intensive aerosol optical properties are sensitive to different aerosol characteristics. The SAE is primarily related to the particle size (Ångström, 1929), with SAE< 1 found for the predominance of coarse aerosol particles and SAE> 2 for the predominance of fine particles (e.g., Schuster et al., 2006; Gyawali et al., 2012). The AAE is mainly linked to the particle composition (e.g., Liu et al., 2018), with AAE $\sim$ 1 for pure black carbon particles, since it absorbs efficiently through the complete



solar spectrum (e.g., Bond, 2001). Higher AAE values correspond to absorbing aerosol particles whose absorption increases

with decreasing wavelength, such as mineral dust and organic matter (called brown carbon) aerosols (e.g., Petzold et al., 2009; Sandradewi et al., 2008; Russell et al., 2010; Valentini et al., 2020). The single scattering albedo Ångström exponent (SSAAE) is sensitive to both aerosol size and composition. Thus, it has been proposed as an additional parameter indicating the presence of mineral dust particles (Collaud Coen et al., 2004). The asymmetry parameter $g$ indicates the asymmetry of the angular distribution of scattered light, varying from 1 in the extreme case of light scattered exclusively in the forward direction to 0

for symmetric scattering. For atmospheric aerosol particles, $g$ is typically in the range of about 0.4 to 0.8 and decreases with decreasing particle size (e.g., Horvath et al., 2018; Pandolfi et al., 2018). The SSA equals 1 if only purely scattering aerosols (e.g., sulfate) are present and decreases as more absorbing aerosols are present. Therefore, $g$ considers the effect of particle sizes, and the SSA takes into account the effect of aerosol absorption on the direct radiative effect.

Several intensive field experiments, including airborne and ground-based measurements, were performed to study the prop-
erties of mineral dust. The main focus has been on the properties of pure mineral dust layers close to the source region and after transport (e.g., Formenti et al., 2003, 2011a; Ryder et al., 2013; Weinzierl et al., 2017; Adebiyi et al., 2023). The most comprehensive field experiment on the properties of Saharan dust was the Saharan Mineral Dust experiment (SAMUM). The first part of the field campaign (SAMUM-1, Heintzenberg, 2009) studied the properties of pure mineral dust in Morocco, whereas SAMUM-2 investigated far-transported, aged dust and as well as its mixture with marine, urban and biomass-burning aerosols,
pointing out the importance of further studies on mixtures of aerosols (Ansmann et al., 2011).

Meanwhile, several studies investigated the feasibility of using the Ångström exponents (SAE, AAE, and SSAAE) to detect mineral dust events in near-real-time using ground based in-situ measurements (e.g., Collaud Coen et al., 2004; Valenzuela et al., 2015; Ealo et al., 2016). They found out that the presence of mixtures my mask the signal of mineral dust particles and only strong mineral dust events can be detected. Therefore, for detecting mineral dust events at a specific emplacement, the
intensive aerosol optical properties need to be evaluated and calibrated with previous sensitivity tests (Ealo et al., 2016).

Studies on mineral dust-pollution mixtures are sparse and conducted mainly in Asia (e.g., Clarke et al., 2004; Seinfeld et al., 2004; Yang et al., 2009), but few field campaigns were also conducted in Western Africa (Petzold et al., 2011; Denjean et al., 2020), and in the western Mediterranean (Mallet et al., 2016; Denjean et al., 2016). Large variability in the obtained results from different studies indicate that internal (Clarke et al., 2004; Quinn et al., 2004) or external (Shi et al., 2003; Song et al., 2005)
mixtures of black carbon and mineral dust may significantly change the optical properties of mineral dust layers. Additionally, different source regions may be affected differently by the mixing with pollution. Petzold et al. (2011) showed that the pollution plume from the city of Dakar had a significant influence on the particle size distribution and absorption properties of mineral dust particles, whereas the absorption Ångström exponent was still dominated by the mineral dust fraction. In Casablanca, a different scenario was observed with an AAE value of $\sim 1$ despite the high mineral dust concentrations Petzold et al. (2009).
In Asia, Seinfeld et al. (2004) found that carbonaceous components in the mineral dust layers changed the SSA from 0.97 to 0.85 at $550\,\mathrm{nm}$. Similarly, in West Africa, Denjean et al. (2020) measured enhanced aerosol light absorption in mineral dust layers due to mixing with pollution, leading to a decrease in SSA decreasing from 0.92 to 0.81. On the contrary, in the western Mediterranean, Denjean et al. (2016) observed moderately absorbing mineral dust layers with SSA values between 0.90 and





1.00 at 530 nm, indicating that the Saharan dust optical properties were not significantly affected by mixing with pollution.
In proximity to Arabian and Saharan dust source regions, Kim et al. (2011) investigated aerosol optical properties, finding no systematic difference in the optical properties of pure Arabian and Saharan dust. Nevertheless, their analysis suggests that the optical properties of mineral dust mixtures may be affected differently depending on the source regions.

In conclusion, mixtures of black carbon and mineral dust may significantly change the optical properties of mineral dust layers. These changes can have consequences on identifying mineral dust layers by using the Ångström exponents (SAE, AAE, and SSAAE) (e.g., Collaud Coen et al., 2004; Valenzuela et al., 2015; Ealo et al., 2016) and on the direct radiative effect by influencing $g$ and SSA (e.g., Seinfeld et al., 2004; Denjean et al., 2020). However, the mixing of pollution with mineral dust from various source regions may also affect the optical properties differently (Kok et al., 2023; Denjean et al., 2016; Seinfeld et al., 2004). Thus, the influence on mineral dust optical properties of mixing with pollution for different source regions needs to be further investigated.

In April 2017, the A-LIFE (Absorbing aerosol layers in a changing climate: aging, lifetime and dynamics, https://www.a-life.at) aircraft field experiment investigated the aerosol properties in the Eastern Mediterranean (Weinzierl et al., in prep.). The region is an atmospheric crossroad and a pollution hot spot (Zittis et al., 2022), where aerosols of different origins (natural vs anthropogenic) co-exist, including mineral dust from the two nearby deserts (the Arabian Peninsula and the Saharan desert) and local and transported pollution (e.g., Dayan et al., 2017). We focus on the intensive aerosol optical properties measured airborne during the A-LIFE field experiment.

The main goal of this paper is to analyze how the mixing of mineral dust and pollution impacts the mineral dust's intensive optical properties and whether these impacts vary depending on the mineral dust's source region. Specifically, we aim to investigate how mineral dust-pollution mixtures affect existing aerosol classification schemes and modify the direct radiative effect of the mineral dust. Thus, we investigate intensive aerosol optical properties that are relevant for the identification of aerosol types, i.e., the scattering, absorption, and single scattering albedo Ångström exponents (SAE, AAE, SSAAE), and relevant for the understanding of the aerosol radiative effect, i.e., the asymmetry parameter and the single scattering albedo ($g$ and SSA). In addition, we used a simple formula by Haywood and Shine (1995) to estimate the change in the short-wave direct radiative effect efficiency (SW DREE) at the top of the atmosphere due to increasing pollution content in the mineral dust layer.



## 2 Methodology

To investigate the influence of mixing mineral dust and pollution on the intensive aerosol optical properties, we analyzed data collected in spring 2017 on board the DLR Falcon 20 E-5 aircraft during the field experiment of the ERC-funded A-LIFE project (Absorbing aerosol layers in a changing climate: aging, lifetime and dynamics, https://www.a-life.at). The particle scattering and absorption coefficients measured directly in Arabian and Saharan dust layers were used to derive intensive aerosol optical properties, including the scattering, absorption and single scattering albedo Ångström exponents (SAE, AAE, and SSAAE), the single scattering albedo (SSA), the asymmetry parameter ($g$), and the short-wave direct radiative effect efficiency (SW DREE). Since data with different relative contribution of mineral dust and black carbon are available, we used the microphysical properties to quantify the pollution content within mineral dust layers. Thus, we investigated the relationships between each intensive aerosol optical property and the aerosol microphysical properties. Additionally, we investigated the differences in these relationships for two different source regions of mineral dust (the Arabian Peninsula and the Saharan desert). The following sections describe the A-LIFE aircraft field experiment, the instrumentation relevant for this study and how the data were analyzed.

### 2.1 The A-LIFE aircraft field experiment in the Eastern Mediterranean

The A-LIFE aircraft field experiment conducted ground-based and airborne measurements with the aim to investigate the properties of absorbing aerosol mixtures, with a focus on mineral dust-black carbon mixtures. The airborne measurements were carried out on board of the Deutsches Zentrum für Luft- und Raumfahrt (DLR) research aircraft Falcon 20-E5, and the aircraft was based at the Paphos airport in Cyprus. This location allowed access to the Eastern Mediterranean region and to mineral dust layers originating from the two nearby deserts, the Arabian Peninsula in the Middle East and the Saharan desert in Northern Africa as well as to anthropogenic pollution (Weinzierl et al., in prep.).

Figure 1 shows the flight paths of the 22 research flights. Apart from the two test flights in Europe and the transfer flights from the DLR airport to the Paphos aircraft base in Cyprus, most of the flights were conducted over the Eastern Mediterranean, close to Cyprus, but also in the area of Crete and Malta. The background of Fig. 1 is a natural color satellite image from April 27, 2017, showing the influence of the regional aerosol loading by mineral dust from the nearby deserts, the Arabian Peninsula and the Saharan desert.

During the field experiment, several outbreaks of Arabian and Saharan dust occurred and the elevated mineral dust layers were also mixed with local and transported anthropogenic pollution, allowing the investigation of mineral dust-pollution mixtures. A detailed description of the field experiment and the meteorological conditions influencing the measurements can be found in the A-LIFE overview paper (Weinzierl et al., in prep.).

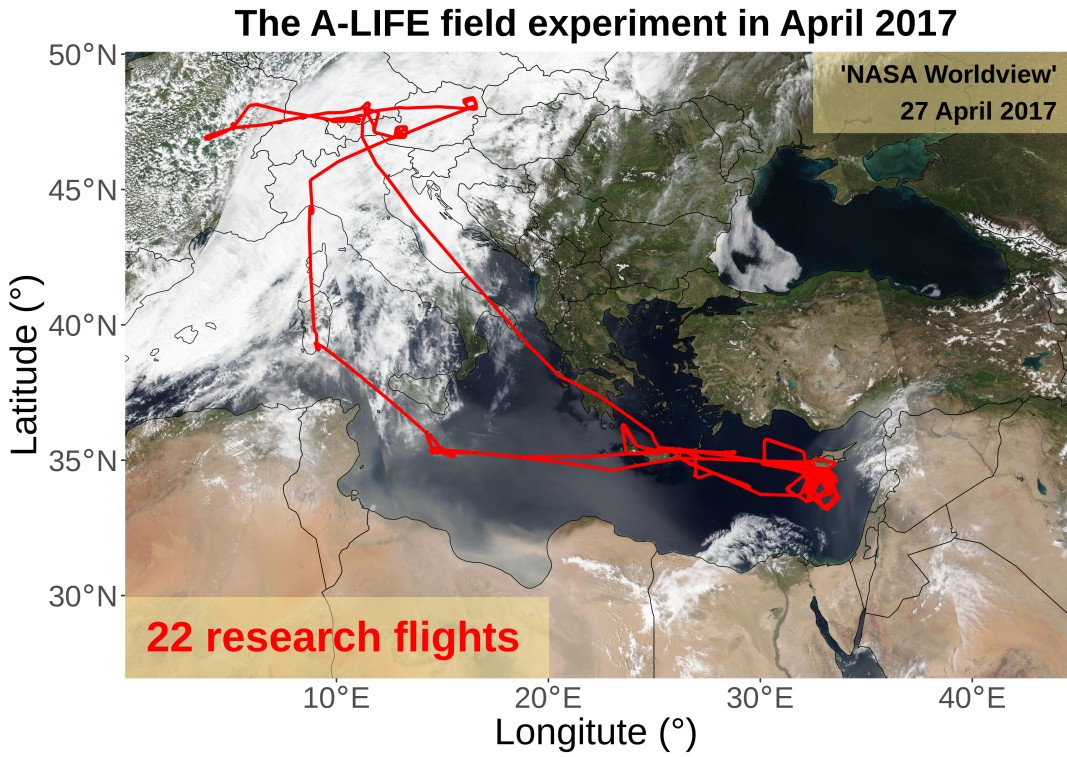

**Figure 1. The A-LIFE field experiment in April 2017.** The red line indicates the flight path of the 22 research flights performed during the A-LIFE field experiment onboard of the Deutsches Zentrum Luft- und Raumfahrt (DLR) research aircraft Falcon 20 E-5. The background is a natural color satellite image (26.9°N-50.1°N, 0.0°E-44.5°E) from 27 April 2017. (Source: "NASA-Worldview", https://worldview.earthdata.nasa.gov/)



## 2.2 Instrumentation

For the A-LIFE field experiment, the DLR aircraft Falcon 20 E-5 was equipped with several in-situ aerosol instruments, a wind lidar, and impactors for subsequent chemical analysis of the filter sample. Most in-situ aerosol instruments were deployed in the Falcon aircraft cabin behind an aerosol inlet and transport system (in-cabin). The cut-off diameter of the Falcon aerosol sampling system ranges between $2.4\,\mu m$ to $6.4\,\mu m$, as characterized in detail in Schöberl et al. (2023). A list of the in-cabin and wing-mounted instruments is available in the supplemental material of (Schöberl et al., 2023).

Instrumentation relevant for this study includes an in-cabin polar integrating nephelometer (Aurora 4000 polar nephelometer, Ecotech, Australia) and an in-cabin Tricolor Absorption Photometer (TAP, Brechtel Manufacturing Inc., Hayward, USA), which measured the aerosol scattering and absorption coefficients at three wavelengths, respectively. An in-cabin Single Particle Soot Photometer (SP2, Droplet Measurement Technologies Inc. (DMT), Longmont, CO, USA) measured the mass and size of individual refractory black carbon particles and provides the mass concentration of refractory black carbon.

The particle size distribution was determined from a combination of instruments with overlapping size ranges: The total number concentration of particles was measured with an in-cabin condensation particle counter (TSI CPC, model 3760a, TSI Inc.). Size resolved-particle number concentration was measured with an in-cabin optical particle counter (SkyOPC, model 1.129, GRIMM Aerosol Technik, Ainring, Germany), and two wing-mounted optical spectrometers, the Ultra-High Sensitivity Aerosol Spectrometer-Airborne (UHSAS-A, DMT), and the Cloud and Aerosol Spectrometer (UNIVIE CAS), which is part 160 of University of Vienna's second-generation Cloud, Aerosol, and Precipitation Spectrometer (UNIVIE CAPS, DMT).

## 2.3 Data analysis

For consistent analysis of the A-LIFE data set, flight time series were divided into 262 sequences, corresponding to flying periods at constant altitude, in the same air mass, and out of clouds (Murphy et al., 2004; Dollner et al., 2023). All data were averaged along these sequences and converted to standard temperature and pressure conditions (STP, 273.15 K; 1013.25 hPa). 165 Data analysis for each instrument is described in the following sections.

### 2.3.1 Directly measured aerosol optical properties

This work focuses on understanding the aerosol optical properties measured directly on board the Falcon research aircraft during A-LIFE by the Aurora 4000 polar nephelometer and the TAP, both installed in-cabin. Therefore, the measured aerosol optical properties are the optical properties of the aerosol passing through the Falcon sampling system. Details on the operation 170 and data analysis of the Aurora 4000 nephelometer and the TAP are reported in the supplementary information (Sections S1 and S2).

The Aurora 4000 polar nephelometer measured the in-cabin particle scattering coefficients ($\sigma_{\mathrm{sp},\lambda}^{\mathrm{Aurora\,4000,\,}\alpha}$) of four angular sectors (from $\alpha$ to $180°$, with $\alpha = 0°, 20°, 50°, 90°$ ) at three wavelengths ($\lambda$): blue ($450\,\mathrm{nm}$), green ($525\,\mathrm{nm}$), and red ($635\,\mathrm{nm}$). Several calibrations with particle-free air and $CO_2$ were performed, and the data were corrected following Ander- 175 son and Ogren (1998). The total particle scattering coefficients ($\sigma_{\mathrm{sp},\lambda}$) were obtained by correcting $\sigma_{\mathrm{sp},\lambda}^{\mathrm{Aurora\,4000,\,}0°}$ for angular





truncation and illumination as indicated in Teri et al. (2022). The overall uncertainty on $\sigma_{\mathrm{sp},\lambda}$ was calculated to be within $\sim 13 - 30\%$ for sequence averages considered in this work.

The TAP measured the in-cabin particle absorption coefficient ($\sigma_{\mathrm{ap},\lambda}$) at three different wavelengths: blue (465 nm), green (520 nm), and red (640 nm). $\sigma_{\mathrm{ap},\lambda}$ was corrected for filter loading and scattering effects with the Virkkula (2010) correction scheme for the widely used glass fiber Pallflex E70-2075W filters. According to Davies et al. (2019), we assumed an uncertainty on $\sigma_{\mathrm{ap},\lambda}$ of $40\%$ for all three wavelengths.

To ensure the accuracy of the data we performed calibrations and data quality controls as reported in the supplementary information (Section S2). In particular, a strong correlation ($R^2 > 0.92$ for all three wavelengths) was found between the particle absorption coefficient and the mass concentration of refractory black carbon (Figure S1 in the supplementary information). The slopes obtained from the linear regression analysis (19.4, 15.9, and $10~\mathrm{m}^2\mathrm{g}^{-1}$ for the blue, green, and red wavelength, respectively) are consistent with the literature values of the black carbon mass absorption coefficient (e.g., Ohata et al., 2021). The slightly higher values obtained from our measurements and the observed wavelength dependence are consistent with the presence of the additional contribution of mineral dust absorption. This comparison indicates good quality of the aerosol absorption coefficient measurements.

### 2.3.2 Intensive aerosol optical properties calculated from direct measurements

Our investigation focused on the intensive aerosol optical properties derived from the Aurora 4000 polar nephelometer and TAP measurements. In particular, we calculated intensive aerosol optical properties that have extensively been used in literature (e.g., Cappa et al., 2016) to distinguish between different aerosol types such as the scattering, absorption and single scattering albedo Ångström exponent (SAE, AAE, SSAE), and intensive aerosol optical properties that are important for understanding and estimating the aerosol direct radiative effect (e.g., Haywood and Shine, 1995; Sherman et al., 2015), such as the asymmetry parameter ($g$) and the single scattering albedo (SSA), as well as the short-wave direct radiative effect efficiency (SW DREE). An overview of the optical properties used in this study is given in table A1 in appendix.

The scattering Ångström exponent (SAE), which represents the wavelength dependence of the particle scattering coefficient (Ångström, 1929), was calculated by using the total particle scattering coefficients $\sigma_{\mathrm{sp},\lambda}$ measured by the Aurora 4000 polar nephelometer at blue and red wavelengths (i.e., $\lambda_{\mathrm{B}} = 450; \mathrm{nm}$ and $\lambda_{\mathrm{R}} = 635; \mathrm{nm}$), as:

$$\mathrm{SAE}_{\lambda_{\mathrm{B}}/\lambda_{\mathrm{R}}} = -\frac{\log(\sigma_{\mathrm{sp},\lambda_{\mathrm{B}}}/\sigma_{\mathrm{sp},\lambda_{\mathrm{R}}})}{\log(\lambda_{\mathrm{B}}/\lambda_{\mathrm{R}})} \tag{1}$$

The absorption Ångström exponent (AAE), which represent the wavelength dependence of the particle absorption coefficient, was calculated by using the particle absorption coefficients $\sigma_{\mathrm{ap},\lambda}$ measured by the TAP at blue and red wavelengths (i.e., $\lambda_{\mathrm{B}} = 465; \mathrm{nm}$ and $\lambda_{\mathrm{R}} = 640; \mathrm{nm}$), as:

$$\mathrm{AAE}_{\lambda_{\mathrm{B}}/\lambda_{\mathrm{R}}} = -\frac{\log(\sigma_{\mathrm{ap},\lambda_{\mathrm{B}}}/\sigma_{\mathrm{ap},\lambda_{\mathrm{R}}})}{\log(\lambda_{\mathrm{B}}/\lambda_{\mathrm{R}})} \tag{2}$$



The single scattering albedo Ångström exponent (SSAAE), which represent the wavelength dependence of the single scattering albedo, was calculated according to Moosmüller and Chakrabarty (2011) as:

$$\mathrm{SSAAE}_{\lambda_\mathrm{B}/\lambda_\mathrm{R}} = \mathrm{SAE}_{\lambda_\mathrm{B}/\lambda_\mathrm{R}} - \mathrm{EAE}_{\lambda_\mathrm{B}/\lambda_\mathrm{R}} \tag{3}$$

where EAE is the extinction Ångström exponent calculated at blue and red wavelengths of the Aurora 4000 polar nephelometer
(i.e., $\lambda_\mathrm{B} = 450\,\mathrm{nm}$ and $\lambda_\mathrm{R} = 635\,\mathrm{nm}$) by using a formula analogous to Eq. 1. The extinction coefficient $\sigma_{ep,\lambda}$ was calculated as the sum of $\sigma_{\mathrm{sp},\lambda}$ and $\sigma_{\mathrm{ap},\lambda}$ measured by the Aurora 4000 polar nephelometer and by the TAP, respectively.

We did not apply any correction for the small difference in the wavelengths of the two instruments, since the centroid wavelengths differ less than the full width at half maximum (FWHM) specified by the manufacturer, and we assumed the correction to be negligible.

The asymmetry parameter $g$, which is the cosine-weighted average of the angular distribution of scattered light (Andrews et al., 2006), was calculated at wavelength $\lambda$ using the particle scattering coefficients measured by the Aurora 4000 polar nephelometer at different angular sectors ($\sigma_{\mathrm{sp},\lambda}^{\text{Aurora 4000, }\alpha}$, with $\alpha = 0°, 20°, 50°, 90°$) following the method developed by Müller et al. (2012):

$$g_\lambda = \frac{\sum\limits_{n}^{N-1}\left(\sigma_{\mathrm{sp},\lambda}^{\text{Aurora 4000, }\alpha_n} - \sigma_{\mathrm{sp},\lambda}^{\text{Aurora 4000, }\alpha_{n+1}}\right)\cos\left(\frac{\alpha_n+\alpha_{n+1}}{2}\right)}{\sum\limits_{n}^{N-1}\left(\sigma_{\mathrm{sp},\lambda}^{\text{Aurora 4000, }\alpha_n} - \sigma_{\mathrm{sp},\lambda}^{\text{Aurora 4000, }\alpha_{n+1}}\right)}1.7752 - 0.6599 \tag{4}$$

The first part of the function is analogous to the definition of the asymmetry parameter. The parameters were found by comparison with the result from scattering theory considering different aerosol types including different size and shapes (Müller et al., 2012).

The single scattering albedo (SSA), which is the fraction of the extinted light that is scattered, was calculated at wavelength $\lambda$ using the total particle scattering coefficient measured by the Aurora 4000 polar nephelometer $\sigma_{\mathrm{sp},\lambda}$ and the particle
absorption coefficient $\sigma_{\mathrm{ap},\lambda}$ measure by the TAP, as:

$$\mathrm{SSA}_\lambda = \frac{\sigma_{\mathrm{sp},\lambda}}{\sigma_{\mathrm{sp},\lambda} + \sigma_{\mathrm{ap},\lambda}} \tag{5}$$

The asymmetry parameter ($g$) and the single scattering albedo (SSA) are key parameters for understanding the aerosol direct radiative effect. We used the $g$ and SSA values measured during A-LIFE for a first estimation of the aerosol direct radiative effect. We used a simple formula derived by Haywood and Shine (1995); Charlson et al. (1991) for calculating the
short-wave aerosol direct radiative effect efficiency (SW DREE) at top-of-atmosphere for an optically thin, partially absorbing atmosphere.[1]

---

[1]Other publications (e.g., Sherman et al., 2015) refer to this quantity as the short-wave direct radiative forcing efficiency (SW DRFE) adopting the same terminology used by the original authors (Haywood and Shine, 1995; Charlson et al., 1991). However, the term „Radiative Forcing" is defined as the TOA net radiative flux change due to the change in a climate change driver relative to a pre-industrial reference period (e.g., the year 1750 in the IPCC report AR5). Thus, we used the term „Radiative Effect", defined as the TOA net radiative flux change resulting from the presence of an atmospheric component.



The SW DREE is the short-wave direct radiative effect (SW DRE) per unit aerosol optical depth (AOD) and is to the first order independent of AOD:

$$\text{SW DREE} = \text{SW DRE}/\text{AOD} = -DS_0 T_{\text{atm}}(1 - A_c)\beta\text{SSA}[(1 - R_{\text{s}})^2 - (2R_{\text{s}}/\text{SSA}\beta)(1 - \text{SSA})] \tag{6}$$

where $D$ is the fractional day length, $S_0$ is the solar constant, $T_{\text{atm}}$ is the atmospheric transmission, $A_c$ is the cloud fraction, $R_{\text{s}}$ is the spectrally averaged surface albedo, SSA is the single scattering albedo, and $\beta$ is the average upscatter fraction, i.e. the fraction of the incident solar radiation that is scattered upward to space. $\beta$ can be obtained from $g$ by using the relationship $\beta = -0.2936g^3 + 0.2556g^2 - 0.4489g + 0.5053$ (Wiscombe and Grams, 1976).

For the non-aerosol-related parameters we used $D = 0.50$, $S_0 = 1370\,\text{W}\text{m}^{-2}$, $T_{\text{atm}} = 0.76$, $A_c = 0.60$, and $R_{\text{s}} = 0.15$ as
suggested by Haywood and Shine (1995). Since the values of $g_\lambda$ and $\text{SSA}_\lambda$ calculated via equations 4 and 5 are the values measured for the aerosol passing through the Falcon sampling system, they are representative for the in cabin NSDs. We applied a correction factor to obtain the values of $g$ and SSA representative for the ambient NSDs ($g_{\lambda,\text{ambient}}$ and $\text{SSA}_{\lambda,\text{ambient}}$). We used $g_{\lambda,\text{ambient}}$ and $\text{SSA}_{\lambda,\text{ambient}}$ at the green wavelength $\lambda = 525\,\text{nm}$ to calculate the SW DREE. A detailed description of the method used to extend the optical properties is in the supplementary information (Section S3).

Although the values obtained from this simplified formula have large uncertainties compared to more advanced radiative transfer calculations, as reported by Yu et al. (2006) and Wendisch et al. (2001), its functional relationship though the SW DREE dependence on SSA and $g$ remain valid. If all others parameters are kept constants, this equation can be used to compare the intrinsic radiative effect efficiency of aerosol at the TOA, as widely done in literature (e.g., Sheridan and Ogren, 1999; Virkkula et al., 2011; Sherman et al., 2015; Shen et al., 2018; Hu et al., 2023).

### 2.3.3 Aerosol microphysical properties

In this work, we use the microphysical aerosol properties measured during A-LIFE on board of the Falcon research aircraft. An overview of the microphysical properties used for the analysis of this study is given in the table A2 in the appendix.

The mass concentration of refractory black carbon ($M_{\text{rBC}}$) was measured by the SP2 installed in the cabin of the DLR Falcon research aircraft. The SP2 calibration was performed with fullerene soot as recommended by Schwarz et al. (2010) and the
detection range for rBC particles was $70 - 980\,\text{nm}$  mass equivalent diameter, assuming void-free spheres with a bulk density of $1.8\,\text{g}/\text{cm}^3$ (Moteki et al., 2010) for rBC particles. Although the uncertainty of the calibration fit was found to be about $10\,\%$, we assumed an $M_{\text{rBC}}$ uncertainty of $25\,\%$ as recommended by Schwarz et al. (2008).

The number concentration of coarse mode aerosol ($N_{\text{coarse}}$) as well as the coarse mode size distribution was measured in the nominal size range $0.5 - 50\,\mu\text{m}$ by the wing-mounted UNIVIE CAS. Details of the calibration and data analysis for this
instrument are provided in Dollner et al. (2023) and Spanu et al. (2020).

For each A-LIFE flight sequence, the number size distribution of the aerosol particles measured behind the inlet system (in-cabin NSD) was derived from a combination of measurements with TSI CPC, UHSAS-A, and SkyOPC and parameterized with a combination of three log-normal distributions. The size distribution of the ambient aerosol (ambient NSD) additionally

 

includes measurements from the UNIVIE CAS and covers the size range from 10 nm to 50 μm. For details see Table A2 in the appendix.

Mass concentrations for various size ranges were derived from these two sets of NSDs by assuming particles with a spherical shape and density based on the modeled aerosol composition. In particular, we calculated the mass concentration of sub$-\mu$m particles ($M_{\text{sub-}\mu m}$), the mass concentration of super$-\mu$m particles passing through the inlet system ($M_{\text{super-}\mu m \text{ in-cabin}}$), and the mass concentration of super$-\mu$m particles including the measurements of the wing-mounted CAS instrument ($M_{\text{super-}\mu m \text{ ambient}}$).

Single particle chemistry analysis was conducted using a scanning electron microscope coupled with energy-dispersive X-ray spectroscopy (SEM-EDX) on in-situ samples collected in the Falcon aircraft cabin behind the aerosol inlet system (Kandler et al., 2011, 2007; Schöberl et al., 2023). The supplementary information (Section S4) provides details on the single particle chemistry analysis and the derived optical properties of mineral dust particles.

### 2.3.4 A-LIFE aerosol classification

To investigate the aerosol properties of different aerosol types, an aerosol classification of all 262 flight sequences was determined. 12 different aerosol types were derived by combining in-situ measurements of coarse mode aerosol particles and refractory black carbon mass with modeled aerosol mass concentrations and compositions. These modeled parameters were obtained by combining source-receptor sensitivities calculated with the Lagrangian dispersion model FLEXPART in backward mode (Stohl et al., 1998; Seibert and Frank, 2004; Pisso et al., 2019) with emission data from the Copernicus Atmospheric Monitoring Service (CAMS; CAMS, 2019; Morcrette et al., 2009). Thus, quantitative information on the type and origin of the measured aerosol is provided along the flight path. Details about the method to derive the modeled aerosol mass concentration and composition are provided in Weinzierl et al. (in prep.).

The A-LIFE aerosol classification scheme divides the data set into flight sequences with (174) and without (88) major contribution of mineral dust in the coarse mode. Mineral dust sequences are further subdivided on the basis of the Arabian or Saharan dust source regions accordingly to the modeled aerosol mass concentration and composition. Sequences without major mineral dust contribution are separated into mixtures with enhanced and low coarse mode contribution. Additional distinctions are made to study the aerosol properties for different pollution content. The mass concentration of refractory black carbon ($M_{\text{rBC}}$) is chosen as a proxy for pollution, and the degree of pollution is established based on the ratio between the coarse mode aerosol number concentration ($N_{\text{coarse}}$) and $M_{\text{rBC}}$. Thus, based on the pollution degree $N_{\text{coarse}}/M_{\text{rBC}}$, each sequence is assigned to a pollution sub-type (pure, moderately-polluted, polluted). The A-LIFE aerosol classification scheme was validated by comparison with the results of single particle chemistry analysis. Details about the A-LIFE aerosol classification can be found in Weinzierl et al. (in prep.).

### 2.3.5 Statistical analysis approach

For this study, the A-LIFE data-set was restricted to sequences in the Eastern Mediterranean with major mineral dust contribution according to the A-LIFE aerosol classification and with available intensive aerosol optical properties. Thus, we investigated





the intensive aerosol optical properties for a total of 87 sequences, among which 37, 40, and 10 sequences are classified as pure mineral dust, moderately-polluted mineral dust, and polluted mineral dust, respectively.

To evaluate the change rate of the intensive aerosol optical properties for increasing pollution content within the mineral dust layer, we investigated the relationships between the intensive aerosol optical properties and the aerosol microphysical

properties. Since the mass concentration of refractory black carbon ($M_{\text{rBC}}$) is a tracer for pollution, we used the ratio between $M_{\text{rBC}}$ and the mass of super-$\mu$m aerosol particles as a measure of the pollution contribution in term of mass within the mineral dust layer. We used the mass of super-$\mu$m aerosol particles that pass the aerosol inlet system ($M_{\text{super-}\mu\text{m in-cabin}}$) for analysis of the change rate of the measured intensive aerosol optical properties because the intensive aerosol optical properties are measured inside the aircraft cabin. For the analysis of change rates of the intensive aerosol optical properties representative for

the ambient NSDs (i.e., $g_{\lambda,\text{ambient}}$, $\text{SSA}_{\lambda,\text{ambient}}$, and SW DREE, see section S3 in the supplementary information), we used the mass of super-$\mu$m aerosol particles in the diameter size range $1-50\,\mu\text{m}$ ($M_{\text{super-}\mu\text{m ambient}}$).

Figure 2 shows the relationship between the pollution content in terms of mass and the A-LIFE classification scheme. Sequences classified as pure mineral dust have low values of $M_{\text{rBC}}/M_{\text{super-}\mu\text{m in-cabin}}$ (see panel (a)) and $M_{\text{rBC}}/M_{\text{super-}\mu\text{m ambient}}$ (see panel (b)). In contrast, both rations have higher values for sequences classified as moderately-polluted and polluted mineral

dust. Thus, the pollution content increases as these ratios increase.

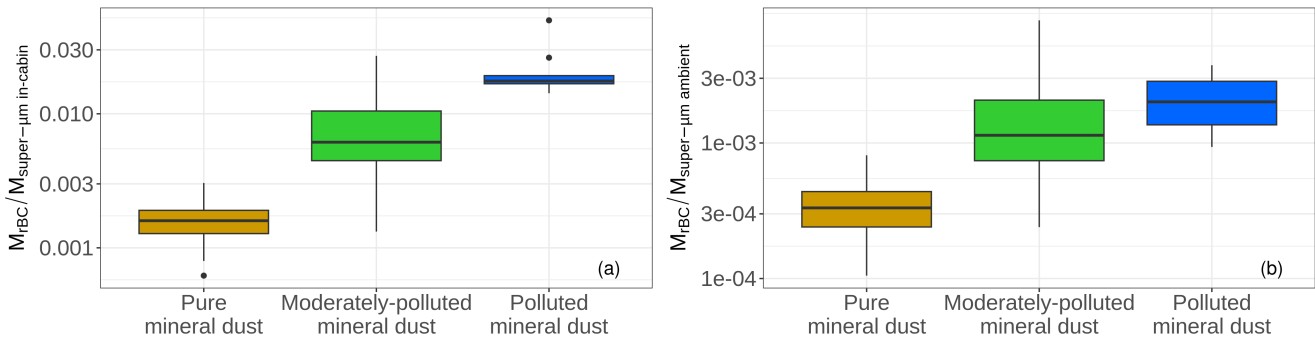

**Figure 2. Box-plot of the pollution contribution in term of mass for the three sub-typespure dust (brown), moderately-polluted dust (green), and polluted dust (blue) according to the A-LIFE aerosol classification Weinzierl et al. (in prep.).** Panel (a) shows the pollution contribution in terms of mass for aerosol particles passing the Falcon aerosol sampling system ($M_{\text{rBC}}/M_{\text{super-}\mu\text{m in-cabin}}$); panel (b) the pollution contribution in terms of mass for the ambient NSD. The boxes indicate the interquartile range (IQR $25^{th}-75^{th}$), the horizontal lines the median, the whiskers the largest value that is not greater than $1.5 \cdot IQR$, and data points outside this range are shown as single dots.

The increase of the pollution content also indicates an increase in the relative mass contribution of fine mode aerosols. Figure 3 shows the comparison between the mass of the sub$-\mu$m aerosols ($M_{\text{sub-}\mu\text{m}}$) and $M_{\text{rBC}}$ in panel (a) and between their contribution relative to the mass of super$-\mu$m aerosol on a log-log scale in panel (b). $M_{\text{sub-}\mu\text{m}}$ is almost 2 orders of magnitude larger than $M_{\text{rBC}}$ indicating the presence of a large number of small particles other than black carbon, e.g., sulfate, nitrates,

organics. Furthermore, their mass concentration increases with increasing $M_{\text{rBC}}$. The high correlation coefficients (r = 0.59



and r = 0.89) indicate the equivalence of investigating the relationship between the intensive aerosol optical properties and the ratios $M_{\text{sub-}\mu\text{m}}/M_{\text{super-}\mu\text{m in-cabin}}$ and $M_{\text{rBC}}/M_{\text{super-}\mu\text{m in-cabin}}$. So we investigated the relationship between each intensive optical property and $log_{10}(M_{\text{rBC}}/M_{\text{super-}\mu\text{m in-cabin}})$. The log transformation is performed due to the skewed distribution of the data.

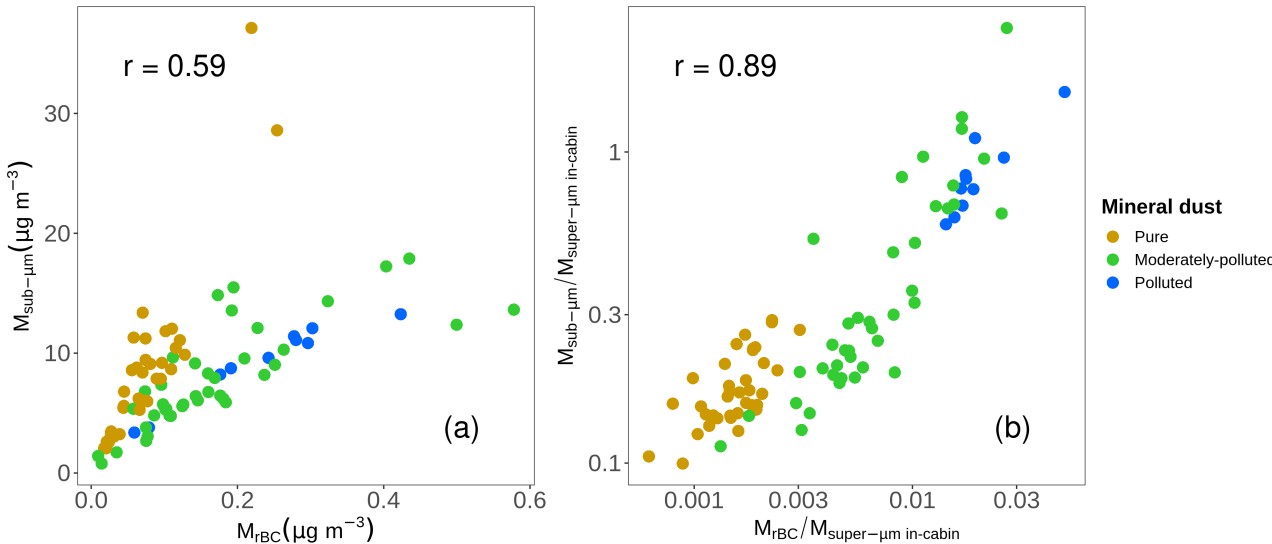

**Figure 3. Mass of sub-$\mu$m particles and mass of refractory black carbon.** Panel (a) shows the comparison between the mass of sub-$\mu$m particles ($M_{\text{sub-}\mu\text{m}}$) and the mass of refractory black carbon ($M_{\text{rBC}}$). Panel (b) shows the comparison on a log-log scale between the contribution of particle size in terms of mass ($M_{\text{sub-}\mu\text{m}}/M_{\text{super-}\mu\text{m in-cabin}}$) and the pollution contribution in terms of mass ($M_{\text{rBC}}/M_{\text{super-}\mu\text{m in-cabin}}$), where $M_{\text{super-}\mu\text{m in-cabin}}$ is the mass of super$-\mu$m particles passing through the aerosol inlet of the DLR Falcon aircraft (Schöberl et al., 2023). Pearson's correlation coefficients **r** are provided on each panel. The color indicates the aerosol types following the A-LIFE aerosol classification scheme.



## 3 Results

To evaluate the impact of mixing pollution and mineral dust from different source regions on the intensive aerosol optical properties, we investigated the intensive aerosol optical properties of sequences measured in-cabin of the Falcon aircraft classified as pure, moderately-polluted, and polluted mineral dust by the A-LIFE aerosol classification scheme (Weinzierl et al., in prep.). Table 1 shows the summary statistics for the intensive aerosol optical properties (SAE, AAE, SSAAE, $g$, and SSA) for pure, moderately polluted, and polluted mineral dust sequences.

In addition, we investigated the relationship between the intensive aerosol optical properties and $M_{\text{rBC}}/M_{\text{super-}\mu\text{m in-cabin}}$, which expresses the pollution contribution in terms of mass in the mineral dust layer. Figure 3 shows each intensive aerosol optical property (SAE, AAE, SSAAE, $g$, SSA, and SW DREE) as a function of the pollution contribution in terms of mass $M_{\text{rBC}}/M_{\text{super-}\mu\text{m in-cabin}}$ on a log-scale. The relationships between the intensive aerosol optical properties and $\log_{10}(M_{\text{rBC}}/M_{\text{super-}\mu\text{m in-cabin}})$ are close to linear. Least-square linear regression analysis was performed for all mineral dust 330 sequences and for the mineral dust from the two source regions (the Arabian Peninsula and the Saharan desert) individually. Results for the linear regression analysis are reported in Table 2.

The Ångström exponents (SAE, AAE, and SSAAE), which are intensive aerosol optical properties often used for identifying aerosol types, change significantly with increasing pollution contribution in terms of mass in the mineral dust layer. The relationships between each Ångström exponent and the pollution contribution in terms of mass are statistically significant, as 335 indicated by the $p-$values well below the threshold of $0.05$. For a unit increase of $\log_{10}(M_{\text{rBC}}/M_{\text{super-}\mu\text{m in-cabin}})$, SAE increases by $1.33 \pm 0.09$, AAE decreases by $2.0 \pm 0.3$, and SSAAE increases by $+0.08 \pm 0.01$. The determination coefficient $R^2$ values suggest that the pollution contribution in terms of mass accounts for $72\%$ of the variability in SAE, while it explains $43\%$, and $30\%$ of the variability in AAE, and SSAAE, respectively.

The asymmetry parameter and single scattering albedo ($g$ and SSA) also change significantly ($p < 0.001$) with increasing 340 pollution contribution in terms of mass in the mineral dust layer. When $\log_{10}(M_{\text{rBC}}/M_{\text{super-}\mu\text{m in-cabin}})$ increases by a unit, $g_{\lambda_{\text{G}}}$ decreases by $0.08 \pm 0.01$ and SSA$_{\lambda_{\text{G}}}$ decreases by $0.079 \pm 0.006$. The pollution contribution in terms of mass is responsible for $49\%$ and $67\%$ of the variability in $g$ and SSA, respectively. Median values of $g$ and SSA measured at the wavelength $\lambda_{\text{G}}$ change from $0.67$ to $0.56$ and from $0.96$ to $= 0.89$, respectively, from pure mineral dust to polluted mineral dust.

For the intensive aerosol optical properties representative for the ambient NSD (see Figure B1 and Table B1 in the ap-345 pendix), the effect of increasing the pollution contribution is also significant, as indicated by the $p$-value $< 0.001$. When $\log_{10}(M_{\text{rBC}}/M_{\text{super-}\mu\text{m in-cabin}})$ increases by a unit, $g_{\lambda_{\text{G,ambient}}}$ decreases by $0.05 \pm 0.01$ and SSA$_{\lambda_{\text{G,ambient}}}$ decreases by $0.044 \pm 0.007$. From pure mineral dust to polluted mineral dust, the median value of SSA$_{\lambda_{\text{G,ambient}}}$ change from $0.93$ to $0.89$; the median value of $g_{\lambda_{\text{G,ambient}}}$ change from $0.71$ to $0.56$.

These results suggest that also the direct radiative effect of mineral dust layers might be affected by the increase of pollution 350 content. For the first estimation of this impact, the short-wavelength direct radiative effect efficiency (SW DREE) was calculated for our data. However, the result of the linear regression analysis between SW DREE and the pollution contribution in terms of mass indicates that the relationship is not statistically significant ($p > 0.05$). The SW DREE increases only slightly by





$0.7 \pm 0.9 W m^{-2} A O D^{-1}$ for a unit increase of $\log_{10}(M_{\text{rBC}}/M_{\text{super-}\mu\text{m ambient}})$. Thus, our data do not provide enough evidence to conclude that the SW DREE changes with increasing pollution content on a global average.

The results for the mineral dust from the two source regions (the Arabian Peninsula and the Saharan desert) are similar. The slopes and intercepts obtained from the linear regression analysis are in agreement within the error bars for almost all intensive aerosol optical properties. Thus, the change rate of the intensive aerosol optical properties (SAE, AAE, SSAAE, $g$, and the SW DREE) for increasing pollution contribution in terms of mass agree within error bars for the mineral dust from the two different source regions. Small differences are found for the SSA. Indeed, the intercept is $0.77 \pm 0.02$ and $0.71 \pm 0.02$ for Arabian and

Saharan dust, respectively. For a unit increase of $\log_{10}(M_{\text{rBC}}/M_{\text{super-}\mu\text{m in-cabin}})$, SSA decreases by $0.068 \pm 0.009$ for Arabian dust and by $0.087 \pm 0.008$ for Saharan dust.

**Table 1.** Summary statistics for each aerosol type of the intensive aerosol optical properties measured inside the cabin of the Falcon research aircraft behind an aerosol sampling system during A-LIFE for measurement sequences classified as pure, moderately-polluted, and polluted mineral dust by the A-LIFE classification scheme.

|  |  | $\text{SAE}_{\lambda_B/\lambda_R}$ | $\text{AAE}_{\lambda_B/\lambda_R}$ | $\text{SSAAE}_{\lambda_B/\lambda_R}$ | $g_{\lambda_B}$ | $g_{\lambda_G}$ | $g_{\lambda_R}$ | $\text{SSA}_{\lambda_B}$ | $\text{SSA}_{\lambda_G}$ | $\text{SSA}_{\lambda_R}$ |
|---|---|---|---|---|---|---|---|---|---|---|
| | median | **-0.02** | **3.50** | **-0.10** | **0.68** | **0.67** | **0.69** | **0.95** | **0.96** | **0.98** |
| | $3^{rd}$ | -0.58 | 0.82 | -0.18 | 0.62 | 0.63 | 0.65 | 0.92 | 0.94 | 0.95 |
| Pure mineral dust | $25^{th}$ | -0.17 | 2.48 | -0.11 | 0.66 | 0.65 | 0.67 | 0.95 | 0.95 | 0.98 |
| | $75^{th}$ | 0.16 | 4.22 | -0.07 | 0.69 | 0.69 | 0.70 | 0.96 | 0.97 | 0.99 |
| | $97^{th}$ | 0.48 | 7.04 | -0.01 | 0.74 | 0.72 | 0.75 | 0.98 | 0.98 | 1.00 |
| | median | **0.58** | **1.92** | **-0.08** | **0.61** | **0.63** | **0.64** | **0.91** | **0.91** | **0.94** |
| | $3^{rd}$ | -0.14 | 0.68 | -0.16 | 0.56 | 0.56 | 0.52 | 0.86 | 0.84 | 0.85 |
| Moderately-polluted mineral dust | $25^{th}$ | 0.30 | 1.52 | -0.11 | 0.59 | 0.59 | 0.61 | 0.89 | 0.89 | 0.91 |
| | $75^{th}$ | 0.97 | 2.36 | -0.05 | 0.64 | 0.65 | 0.66 | 0.93 | 0.93 | 0.96 |
| | $97^{th}$ | 2.18 | 2.79 | 0.11 | 0.67 | 0.70 | 0.69 | 0.95 | 0.95 | 0.97 |
| | median | **1.73** | **1.70** | **0.01** | **0.56** | **0.56** | **0.54** | **0.89** | **0.89** | **0.88** |
| | $3^{rd}$ | 1.29 | 1.48 | -0.05 | 0.49 | 0.50 | 0.50 | 0.86 | 0.84 | 0.85 |
| Polluted mineral dust | $25^{th}$ | 1.57 | 1.65 | -0.00 | 0.55 | 0.54 | 0.52 | 0.88 | 0.87 | 0.88 |
| | $75^{th}$ | 1.89 | 1.75 | 0.02 | 0.57 | 0.57 | 0.56 | 0.90 | 0.89 | 0.90 |
| | $97^{th}$ | 2.02 | 2.16 | 0.08 | 0.58 | 0.59 | 0.58 | 0.95 | 0.94 | 0.95 |







**Figure 4. Intensive aerosol optical properties of mixtures of pollution and mineral dust from different source origin.** Each panel shows a different intensive optical property as a function of the pollution contribution in terms of mass ($M_{\text{rBC}}/M_{\text{super-}\mu\text{m in-cabin}}$). Panel **(a)** shows the wavelength dependence of the scattering coefficient at $\lambda = 450$, and 635 nm (SAE$_{\lambda_{\text{B}}/\lambda_{\text{R}}}$), panel **(b)** shows the wavelength dependence of the absorption coefficient at $\lambda = 465$, and 640 nm (AAE$_{\lambda_{\text{B}}/\lambda_{\text{R}}}$), panel **(c)** shows the wavelength dependence of the single scattering albedo (SSAAE$_{\lambda_{\text{B}}/\lambda_{\text{R}}}$), panel **(d)** shows the asymmetry parameter ($g$) at $\lambda = 525$ nm, panel **(e)** shows the single scattering albedo (SSA) at $\lambda = 525$ nm, and panel **(f)** shows the short-wave direct radiative effect efficiency (SW DREE) calculated with a simplified formula (Haywood and Shine, 1995), the measured values of $g$ and SSA at $\lambda = 525$ nm and global average parameters. $g$ and SSA were extended to the ambient particle size distribution (see section S3 in the supplementary information). Thus, the SW DREE is plotted as function of $M_{\text{rBC}}/M_{\text{super-}\mu\text{m ambient}}$), where $M_{\text{super-}\mu\text{m ambient}}$ is the mass of the ambient super$-\mu$m particles.

Only sequences classified as mineral dust by the A-LIFE aerosol classification are shown. The color code indicates the mineral dust type. Least squares regression results, including regression line and equation, determination coefficient and $p-$value, are shown in the corresponding mineral dust type color.





**Table 2.** Results of the linear regression analysis of each intensive aerosol optical property vs. the pollution contribution in terms of mass $log_{10}(M_{\text{rBC}}/M_{\text{super-}\mu\text{m in-cabin}})$. Data used to perform the linear regression analysis are plotted in Figure 4.

| | | $\text{SAE}_{\lambda_B/\lambda_R}$ | $\text{AAE}_{\lambda_B/\lambda_R}$ | $\text{SSAAE}_{\lambda_B/\lambda_R}$ | $g_{\lambda_G}$ | $\text{SSA}_{\lambda_G}$ | SW DREE |
|---|---|---|---|---|---|---|---|
| | slope | +1.33 ($\pm$ 0.09) | -2.0 ($\pm$ 0.3) | +0.08 ($\pm$ 0.01) | -0.08 ($\pm$ 0.01) | -0.079 ($\pm$ 0.006) | +0.7 ($\pm$ 0.9) |
| | intercept | +3.7($\pm$ 0.2) | -2.3 ($\pm$ 0.6) | +0.12($\pm$ 0.03) | +0.45 ($\pm$ 0.02) | +0.74 ($\pm$ 0.01) | -16 ($\pm$ 3) |
| All | $R^2$ | 0.72 | 0.43 | 0.3 | 0.49 | 0.67 | 0.01 |
| | significance | $p < 0.001$ | $p < 0.001$ | $p < 0.001$ | $p < 0.001$ | $p < 0.001$ | $p = 0.4$ |
| | slope | +1.4 ($\pm$ 0.1) | -2.3 ($\pm$ 0.4) | +0.10 ($\pm$ 0.02) | -0.06 ($\pm$ 0.01) | -0.068 ($\pm$ 0.009) | +1 ($\pm$ 1) |
| | intercept | +4.0($\pm$ 0.3) | -2.4 ($\pm$ 0.8) | +0.16 ($\pm$ 0.05) | +0.45 ($\pm$ 0.03) | +0.77 ($\pm$ 0.02) | -16 ($\pm$ 4) |
| Arabian dust | $R^2$ | 0.84 | 0.68 | 0.53 | 0.58 | 0.75 | 0.02 |
| | significance | $p < 0.001$ | $p < 0.001$ | $p < 0.001$ | $p < 0.001$ | $p < 0.001$ | $p = 0.56$ |
| | slope | +1.2 ($\pm$ 0.1) | -2.2 ($\pm$ 0.3) | +0.07 ($\pm$ 0.02) | -0.07 ($\pm$ 0.01) | -0.087 ($\pm$ 0.008) | +1 ($\pm$ 1) |
| | intercept | +3.4($\pm$ 0.3) | -2.7 ($\pm$ 0.9) | +0.09($\pm$ 0.04) | +0.48 ($\pm$ 0.03) | +0.71 ($\pm$ 0.02) | -14 ($\pm$ 4) |
| Saharan dust | $R^2$ | 0.63 | 0.38 | 0.19 | 0.41 | 0.65 | 0.02 |
| | significance | $p < 0.001$ | $p < 0.001$ | $p < 0.001$ | $p < 0.001$ | $p < 0.001$ | $p = 0.28$ |





## 4 Discussion

### 4.1 Comparison of mineral dust's optical properties from Arabian and Saharan source regions

Differences in the optical properties of mineral dust particles from different source regions can arise due to differences in
the mineralogical compositions (Sokolik and Toon, 1999; Di Biagio et al., 2019), but also due to differences in the mixing
processes with pollution (Kok et al., 2023; Denjean et al., 2016; Seinfeld et al., 2004). We investigated the differences in
mineral dust layers' intensive aerosol optical properties (SAE, AAE, SSAAE, $g$, SSA, and SW DREE) from two different
source regions (the Arabian Peninsula and the Saharan desert) in the Eastern Mediterranean. We found that the intensive
aerosol optical properties change significantly with the increase of pollution content within the mineral dust layer. On the
contrary, the differences between Arabian and Saharan dust are mostly not significant (see Figure 4) for all intensive aerosol
optical properties except for the SSA, where minor differences are found. These results suggest that there is no major intrinsic
difference in the mineral dust from Arabian and Saharan dust source regions that causes a significant difference in the intensive
aerosol optical properties.

Our findings align with observations of other in-situ aerosol properties measured during the A-LIFE field experiment such as
the single particle chemistry analysis and the particle volume size distributions, and are also consistent with previous research.
For instance, Kim et al. (2011) analyzed the mineral dust optical properties using sun photometers at 14 AERONET sites. Their
study found no systematic difference in the Ångström exponent, single scattering albedo and asymmetry parameter for pure
mineral dust layers from the Arabian and Saharan source regions.

Single particle chemistry analysis of mineral dust particles reveals comparable features with only slight variations between
Arabian and Saharan dust samples (see Figure S3 and S4 in the supplementary information). Both exhibit a high percentage
($\sim 60 - 62\%$) of silicate-like particles, while Saharan dust samples contain more clay-like minerals (Illite and Kaolinite), and
Arabian dust samples show higher carbonate content. These observations align with previous Saharan dust studies (Kandler
et al., 2009; Formenti et al., 2011b). Limited studies on Arabian dust (Attiya and Jones, 2020; Engelbrecht et al., 2009, 2016)
reveal significant variability in mineralogy across the Arabian Peninsula. Thus, identifying specific characteristics of Arabian
dust remains challenging, but studies consistently note higher carbonate and lower clay content compared to Saharan dust.

Our observations suggest that Saharan dust particles were slightly more absorbing than Arabian dust particles over the
Eastern Mediterranean. The linear regression analysis intercepts for directly measured SSA were $0.77 \pm 0.02$ and $0.71 \pm 0.02$
for Arabian and Saharan dust, respectively (see Figure 4(f)). This outcome aligns with the small difference found in the
amount of iron oxides and hydroxides obtained from single particle chemistry analysis. Saharan dust contains about $3\%$ these
compounds, whereas Arabian dust samples exhibit roughly half that amount. This difference results in minor variations in the
imaginary part of the refractive index, as shown in Figure S5 (supplementary information). For instance, the imaginary part
of the refractive index, at the green wavelength $520\,\mathrm{nm}$, has a value of 0.006 for Saharan dust and of 0.005 for Arabian dust.
Consequently, the calculated SSA values of mineral dust particles indicate a $4\%$ increase for Arabian dust compared to Saharan
dust.



Nevertheless, the observed imaginary refractive index values are higher than those of Saharan dust measured in other lo-cations, such as over the Northwest Sahara or the Atlantic (Kandler et al., 2011, 2007). Thus, the observed difference in the absorption properties of Saharan and Arabian dust could be attributed to the different sub-regions of the Saharan desert and the Arabian Peninsula, which may vary for different seasons and their combination in the Eastern Mediterranean.

    Measurements of the particle volume size distribution also do not show a major difference for Arabian and Saharan dust
sequences. However, the particle volume size distributions normalized by the full volume for Arabian dust sequences falls in the larger diameter range of the Saharan dust distributions. This result might be due to the fact that Saharan dust traveled different distances across the Mediterranean depending on the sub-source regions of the Saharan desert. Indeed, coarse mode particles settle due to gravity and the particle size distribution changes during transport, as shown after transport of Saharan dust across the Atlantic by Weinzierl et al. (2017). Nevertheless, differences in the upper diameter range of the size distribution
would not affect the intensive aerosol optical properties measured in the Falcon aircraft cabin (in-cabin) due to the aerosol sampling system efficiency.

    On the contrary, differences between Arabian and Saharan dust were observed in the lidar ratio by several past studies (e.g, Schuster et al., 2012; Mamouri et al., 2016; Nisantzi et al., 2015), as well as during the A-LIFE field experiment (Groß et al., 2024). Higher values are reported for the lidar ratio of Saharan dust compared to the one of Arabian dust. The lidar ratio is an
important intensive aerosol optical property in the framework of aerosol typing effort. It is obtained as the ratio of the extinction coefficient and the back-scattering coefficient at $180°$ measured by a lidar, and it is sensitive to the particle composition, particle size distribution, and particle shape. According to Groß et al. (2013), the lidar ratio decreases with increasing particle size and increases with increasing light absorption. In addition, the lidar ratio decreases with increasing real part of the refractive index (Gasteiger et al., 2011).

The reason for the differences in Arabian and Saharan dust's lidar ratio is unknown. Most of the authors attribute the lidar ratio difference to the different geochemical characteristics of the source region soil, with different ilite concentrations for the two source regions, which influence the real part of the refractive index (e.g., Schuster et al., 2012; Filioglou et al., 2020). However, the lidar ratio can be influenced for various reasons. For example, Hu et al. (2020) pointed out that the lidar ratios can also be influenced by the presence of coarse-mode and giant (d$> 40\mu m$) particles, for example, when measured close to the
source region. Nisantzi et al. (2015) acknowledges that the retrieval of the mineral dust lidar ratio may be affected by pollution if present in the boundary layer or mixed into the mineral dust layer.

    Understanding the reason for the differences in the lidar ratios of Arabian and Saharan dust is out of the scope of this paper. However, we highlight that care is needed when retrieving the optical properties of pure mineral dust because they may contain mixtures of other components, such as black carbon or other small anthropogenic particles. In particular, when analyzing
the differences between different dust types, evaluating the pollution content and comparing only measurements with similar pollution content is essential. While the pollution significantly changes the optical properties, we can not conclude the same for the mineral dust from Arabian and Saharan dust source regions.




## 4.2 Implications for the identification of aerosol types

The scattering, absorption and single scattering albedo Ångström exponents (SAE, AAE, and SSAAE) are sensitive to different
aerosol characteristics. Thus, different aerosol classification schemes deploying these properties exist in literature (e.g., Russell
et al., 2010; Lee et al., 2012; Costabile et al., 2013; Cazorla et al., 2013; Cappa et al., 2016). Most of these studies agree that
high SAE values suggest the presence of fine mode aerosols (e.g., BC, sulfates, or nitrates) and low SAE values indicate large
aerosols (e.g., sea salt and mineral dust). Furthermore, and that AAE values around 1 are associated with BC and higher AAE
values suggest the presence of mineral dust and/or brown carbon. In addition, negative SSAAE values have been associated
with mineral dust outbreaks (e.g., Collaud Coen et al., 2004). However, different studies report slightly different threshold
values for different aerosol types, likely due to differences in the particle size distribution measured at different sites and
with different techniques and to different wavelength pairs used to calculate the intensive aerosol optical properties. Different
classification schemes and the respective threshold values are reviewed in Schmeisser et al. (2017) and Valentini et al. (2020).

We compared our measurements with the classification scheme proposed by Cappa et al. (2016). Figure 5 shows the clas-
sification matrix (AAE vs. SAE) for the sequences classified as mineral dust by the A-LIFE classification scheme (Weinzierl
et al., in prep.). In the upper panel, the points are color-coded by the SSAAE. In the lower panel, the color indicates the aerosol
type according to the A-LIFE classification scheme (Weinzierl et al., in prep.).

For pure mineral dust sequences, our measurements align with observations of previous studies. Measured median values
for pure mineral dust sequences are $-0.02$, $3.50$, and $-0.10$ for SAE, AAE, and SSAAE, respectively (see Table 1). Most
sequences classified as pure mineral dust fall in the "Dust dominated region" of the AAE vs. SAE plot according to the
classification scheme proposed by Cappa et al. (2016) (SAE < 0.5, and AAE > 2). In addition, all pure mineral dust cases have
SSAAE < 0, consistently with values reported by Collaud Coen et al. (2004) and Kaskaoutis et al. (2021).

However, our results show that the Ångström exponents change significantly for increasing pollution content within the
mineral dust layer (Figure 4 and Table 2). Median values for polluted mineral dust sequences are $1.73$, $1.70$, and $0.01$ for SAE,
AAE, and SSAAE, respectively (see Table 1). These values are more similar to values measured for anthropogenic aerosols
with a strong BC component (e.g., Lee et al., 2012). Indeed, polluted mineral dust sequences are mainly in the "Mixed BC/BrC
region" of the AAE vs. SAE accordingly to the Cappa et al. (2016) classification scheme (Figure 5). Moderately-polluted
mineral dust sequences have SAE, AAE, and SSAAE values with a larger variability. Thus, these sequences fall in different
regions according to the Cappa et al. (2016) classification scheme, including "Dust dominated", "Mixed Dust/BC/BrC", and
"BC dominated" regions. Thus, mineral dust can be present for several combinations of the optical properties SAE, AAE, and
SSAAE.

This result indicates that the Ångström exponents (SAE, AAE, and SSAAE) should not be used as indicators to reveal the
presence of mineral dust because the optical properties shift towards values associated with anthropogenic aerosols as the
presence of pollution increases within the mineral dust layer. A few authors already pointed out that the presence of pollution
may mask the mineral dust signal (Esteve et al., 2012; Ealo et al., 2016; Pandolfi et al., 2018). Therefore, independent methods
are needed to identify mineral dust events in mixtures with pollution.



Once the presence of mineral dust is already elucidated, the Ångström exponents (SAE, AAE, and SSAAE) can be useful parameters to estimate the amount of pollution mixed into the mineral dust layer. Indeed, a relationship between all the three Ångström exponents (SAE, AAE, and SSAAE) and the pollution contribution in terms of mass exists (Figure 4, Table 2).

Therefore, the Ångström exponents (SAE, AAE, and SSAAE) can be used to select cases where non-dust aerosols have a minimal impact on the aerosol optical properties and study the properties of pure mineral dust. For example, Müller et al. (2011); Horvath et al. (2018); Kim et al. (2011); Tian et al. (2018) used SAE and or AAE to select pure mineral dust cases and study other optical and microphysical properties. In addition, among the three Ångström exponents (SAE, AAE, and SSAAE), the pollution contribution in terms of mass seems to better account for the variability in SAE (72%), while for AAE and

SSAAE, the pollution contribution in terms of mass explains 43% and 30% of the variability (see Table 2). This result suggests that SAE might be the most reliable intensive aerosol optical property for estimating the amount of pollution in a mineral dust layer. However, it is crucial to confirm the presence of mineral dust using additional, independent methods.



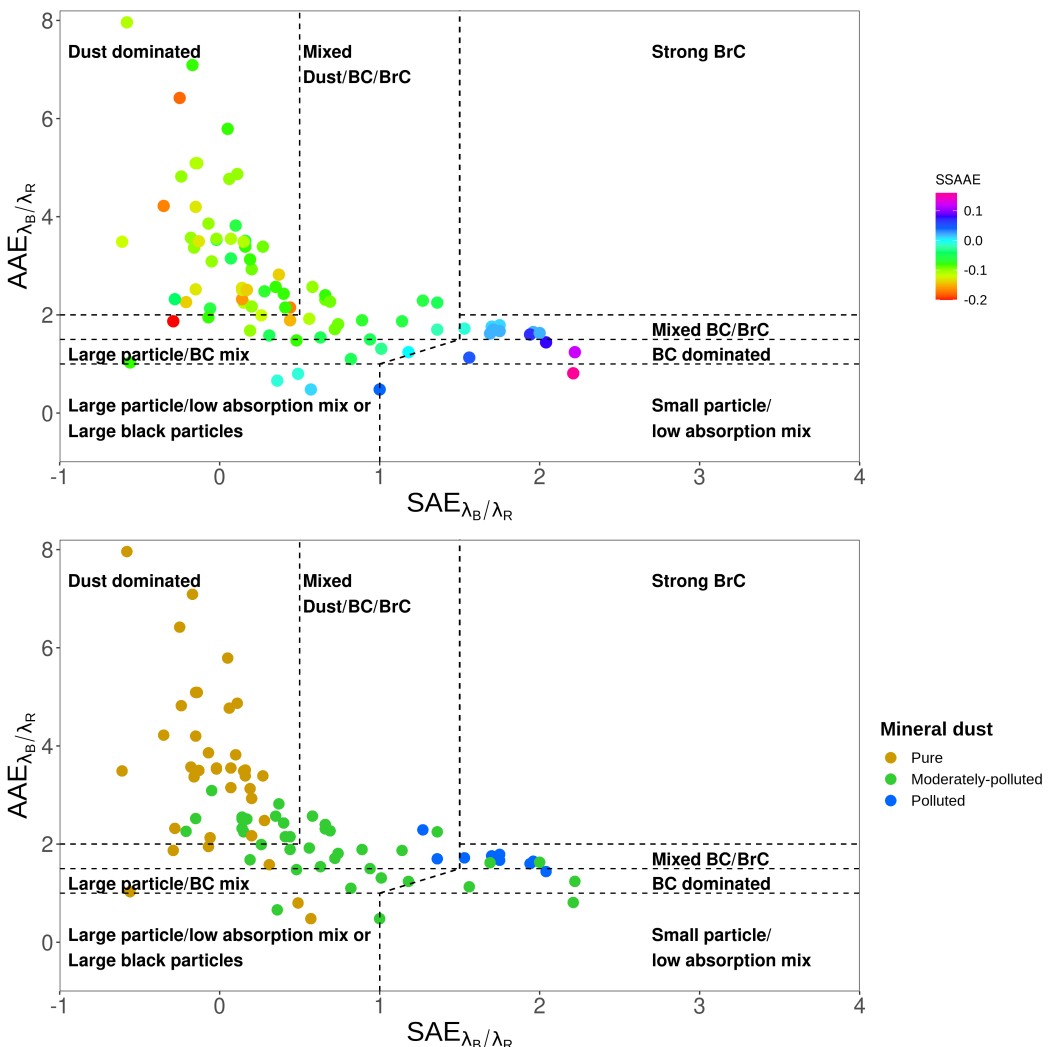

**Figure 5. Aerosol classification schemes using intensive aerosol optical properties.** Both panels show the absorption vs. scattering Ångström exponent matrix ($AAE_{\lambda_B/\lambda_R}$ vs. $SAE_{\lambda_B/\lambda_R}$) overlaid with the Cappa et al. (2016) aerosol classification scheme: aerosol types are reported in black, and the thresholds are indicated with dashed black lines. In the upper panel, the color code indicates the single scattering albedo Ångström exponent ($SSAAE_{\lambda_B/\lambda_R}$). In the lower panel, the color indicates the aerosol types following the A-LIFE aerosol classification scheme.



### 4.3 Asymmetry parameter and single scattering albedo of pure mineral dust layers

The asymmetry parameter ($g$) and the single scattering albedo (SSA) are key intensive aerosol optical properties to understand
the direct radiative effect of mineral dust. The large variability of these optical properties for mineral dust in the atmosphere
may be due to differences in the mineralogical composition and mixture with other aerosol types. Still, it can also arise from
differences in the measurement methods or retrieval technique and the measured particle diameter size range.

For pure mineral dust layers, the values of $g$ and SSA measured during A-LIFE are mainly consistent with literature values
measured during different field campaigns. We consider the values of $g$ and SSA measured in the aircraft cabin representative
of the in-cabin NSDs, as well as the values obtained extending these values to the ambient NSDs (10 nm - 50 μm), which are
thus representative of the ambient NSD.

Figures 6 show the spectral values of $g$ for sequences classified as pure mineral dust directly measured in-cabin of the Falcon
aircraft and extended to the ambient NSD, as well as, values from other field campaigns. Values of $g$ from our measurements are
in the range of values from other measurement campaigns, and a better agreement is shown for values extended to the ambient
NSD (Figure 6). The value of $g$ at the green wavelength (525 nm) extended to the ambient size distribution ($10\,\text{nm} - 50\,\text{μm}$) is
in excellent agreement with the value ($g = 0.71 \pm 0.03$) measured directly with a polar nephelometer by Horvath et al. (2018)
in Sierra Nevada (Spain). Values of $g$ for pure mineral dust obtained from other aircraft measurements performed close to
Saharan dust source regions (e.g., Formenti et al., 2011a; Ryder et al., 2018; Denjean et al., 2020) or after transport through
the Mediterranean (Denjean et al., 2016) agree with our measurements values extended to the ambient NSD. These studies
calculated the intensive optical properties through an inversion closure study combining the particle number size distribution,
direct measurements of the optical properties and the Mie theory.

Figure 7 shows the spectral single scattering albedo (SSA) for pure mineral dust layers in this work in comparison with the
results of other studies. Our measurements of SSA are consistent with the most values from other measurement campaigns
using inversion closure study (e.g., Formenti et al., 2011a; Ryder et al., 2018; Denjean et al., 2016), but also from direct
measurements (e.g., Schladitz et al., 2009; Müller et al., 2011). However, most mean values from other campaigns fall within
the range between the values for our in-cabin and ambient SSA values. For example, the SSA at $\lambda = 525\,\text{nm}$ changes from a
value measured in-cabin of $0.96 \pm 0.02$ to a value representative for the ambient NSD of $0.93 \pm 0.03$ (see Tables 1 and B1).
Ryder et al. (2013) reported optical properties measured inside the aircraft cabin and mie-simulated from the number particle
size distribution in the range $0.1 - 930\,\text{μm}$, and show that the direct measurements overestimate the SSA by up to 0.11 due to
inlet efficiencies. Ryder et al. (2018) showed comparable outcomes, suggesting that the variability in the observed SSA values
may be also due to discrepancies in the measured diameter size range of different studies.

The values reported by Denjean et al. (2020) deviate significantly from our measurements but still are within the variability
range. Their intensive aerosol optical properties were derived from an optical closure study combining size distributions and
optical properties measured for an aerosol particle passing through a PM$_5$ isokinetic inlet. This difference might be due also to
the unidentified presence of pollution mixed with the mineral dust.




Differences in the observed intensive aerosol optical properties ($g$ and SSA) of pure mineral dust layers may be due to variations in the measured particle diameter size range, distinct measurement or retrieval techniques, and different methods used to classify pure mineral dust layers.

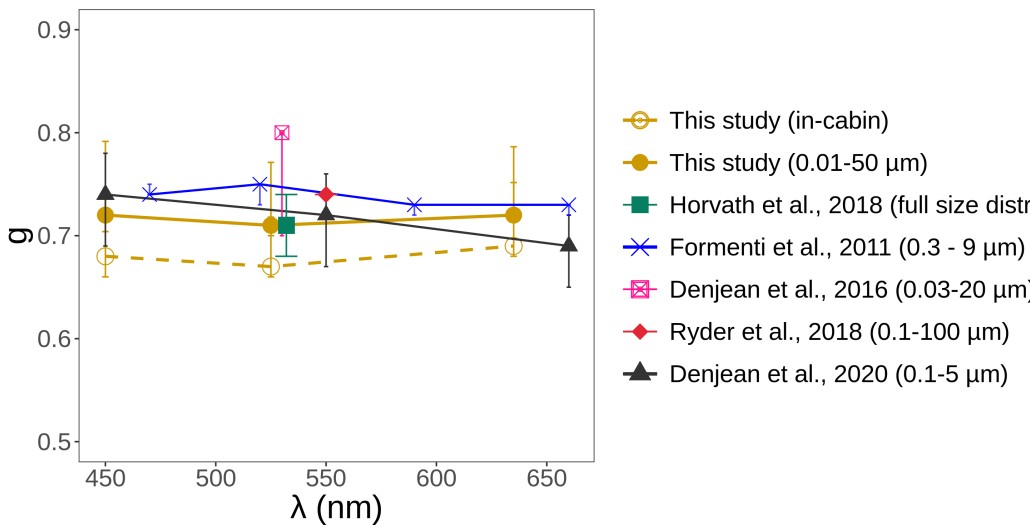

**Figure 6. Spectral asymmetry parameter ($g$) for pure mineral dust layers.** Data from this study are reported in brown. Empty circles indicate values measured in-cabin of the Falcon aircraft with the Aurora 4000 polar nephelometer, while full circles indicate values extended to the full-size distribution. Error bars indicate the variability obtained as the $3^{rd}$ and $97^{th}$ percentiles. For comparison, values issued from the AMMA (Formenti et al., 2011a), CHARMEX/ADRIMED (Denjean et al., 2016), AER-D (Ryder et al., 2018), and DACCIWA (Denjean et al., 2020) field campaigns, as well as direct measurements performed with a polar nephelometer in Sierra Nevada (Spain) by Horvath et al. (2018) are shown. The measured diameter size range for each field experiment is given in the legend. Error bars correspond to the range of variability reported in the publications from which data are taken.





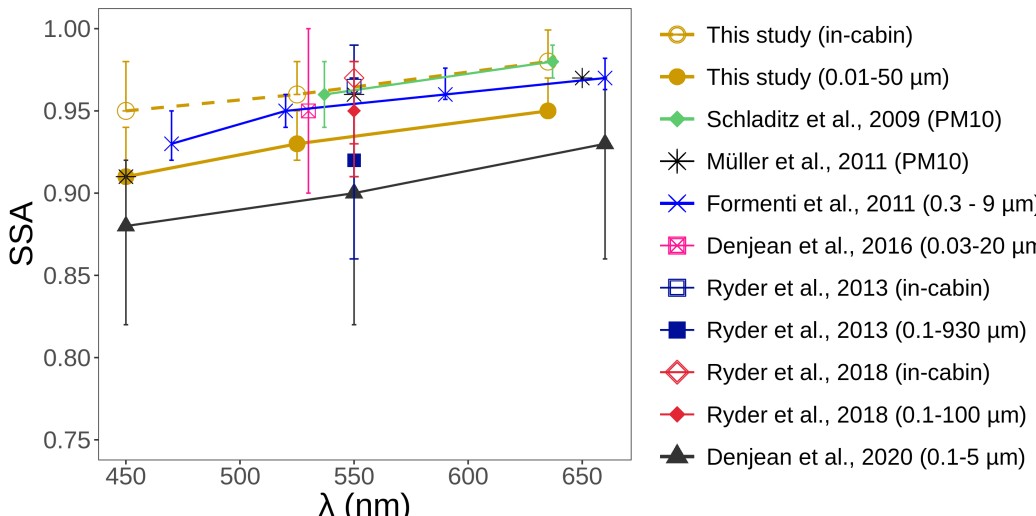

**Figure 7. Spectral single scattering albedo (SSA) for pure mineral dust layers.** Data from this study are reported in brown. Empty circles indicate values measured in-cabin of the falcon aircraft with the Aurora 4000 polar nephelometer and TAP, while full circles indicate values extended to the full-size distribution. Error bars indicate the variability obtained as the $3^{rd}$ and $97^{th}$ percentiles. For comparison, values issued from the SAMUM (Schladitz et al., 2009), SAMUM-2 (Müller et al., 2011), AMMA (Formenti et al., 2011a), CHARMEX/ADRIMED (Denjean et al., 2016), Fennec 2011 (Ryder et al., 2013), AER-D (Ryder et al., 2018) and DACCIWA (Denjean et al., 2020) field campaigns are shown. The measured diameter size range for each field experiment is given in the legend. Error bars correspond to the range of variability reported in the publications from which data are taken.





## 4.4 Implications for the direct radiative effect

The asymmetry parameter ($g$) and the single scattering albedo (SSA) are two intensive aerosol optical properties essential for estimating the aerosol direct radiative effect. While the asymmetry parameter considers the effect of particle size, the single scattering albedo considers the particle absorption's effect.

Our results show that both properties $g$ and SSA decrease significantly for increasing pollution content within the mineral dust layer (Figure 4). $g$ median values change from 0.67 to 0.56 for pure to polluted mineral dust at 525 nm, while SSA median

values change from 0.96 to 0.89 (Table 1). Similar results were obtained, for example, by Seinfeld et al. (2004) in Asia and by Denjean et al. (2020) in West Africa. However, slightly different numbers can be due to the different diameter size ranges measured and the different methods to identify the presence of pollution.

Since the changes in $g$ and SSA are significant with increasing pollution within a mineral dust layer, the direct radiative effect efficiency may also be significantly affected. Indeed, a decrease in $g$ indicates that less light is scattered in the forward

direction, less energy reaches the Earth's surface, and thus the cooling effect may be enhanced compared to a pure mineral dust layer with the same aerosol optical depth (AOD). Conversely, a decrease in SSA indicates that more energy is absorbed in the aerosol layer. Thus, the cooling effect may be reduced (or the warming effect may be enhanced) compared to a pure mineral dust layer with the same AOD. These two opposing effects are due to the reduced particle size and the increased black carbon concentration associated with the increased pollution in the mineral dust layer. Indeed, the pollution includes not only only

black carbon particles, but also fine mode anthropogenic particles (see Figure 3).

The simple formula, developed by Charlson et al. (1991) and Haywood and Shine (1995), can be used to evaluate the impact of the mixing of pollution and mineral dust on the short-wave direct radiative effect efficiency (SW DREE) at the top of the atmosphere by investigating which of the two opposing effect (decrease in particle size vs. decrease in absorption) dominates in our measurements.

Several studies highlighted the limitation of this formula (e.g., Yu et al., 2006). First, the aerosol properties are assumed for a single wavelength (e.g., 525 nm) representative of the entire solar spectrum. Second, multiple scattering effects are neglected, and this formula is adequate only for thin atmospheres (Haywood and Shine, 1995). Third, the vertical profile of constituents is not considered. Fourth, the dependence on the solar zenith angle is not considered. Wendisch et al. (2001) showed that these simplifications caused large uncertainties, especially neglecting the multiple scattering. Hassan et al. (2015) compared

this equation with the output of a more sophisticated model and found it erroneous for all-sky radiative effect but adequate for cloud-free conditions. Thus, despite the large uncertainties, the functional relationship between the SW DREE, $g$ and SSA remains valid. In conclusion, the Haywood and Shine (1995) formula can be used to compare the intrinsic forcing efficiency of different aerosols, as used for example by Sherman et al. (2015). Therefore, we focus not on the absolute number obtained with this formula but on its change rate for increasing pollution within the mineral dust layer. We believe that important conclusions

might be drawn by considering the slopes of the relationship between the SW DREE and the pollution contribution in terms of mass in a mineral dust layer.



On a global average, the pollution increase in a mineral dust layer does not seem to affect the short-wave direct radiative effect of the aerosol layer. Our results (Table 2) show that the SW DREE, calculated considering global average surface albedo and cloud fraction ($R_\mathrm{s} = 0.15$ and $A_\mathrm{c} = 0.60$ ), increases by $0.7(\pm 0.9) W m^{-2} AOD^{-1}$ per unit change of

$\log_{10}(M_\mathrm{rBC}/M_{\text{super-}\mu\text{m ambient}})$. This change is not statistically significant given the wide confidence interval and the high $p-$value of $0.4$. This result suggests that the direct radiative effect of a mineral dust layer with more pollution content does not change compared to a pure mineral dust layer with the same AOD because the decreased size and increased absorption counteract each other.

However, the direct radiative effect of a mineral dust layer can strongly differ depending on the underlying surface albedo

and also on the fraction of sky covered by clouds. The simple formula from Haywood and Shine (1995) (Eq. 6) allows to calculate the SW DREE taking into account for variation of the surface albedo $R_\mathrm{s}$ and of the cloud fraction $A_\mathrm{c}$. For the latter, only the case of clouds above the aerosol layer , i.e., no aerosol effect when the sky is covered by clouds, is considered.

We considered different surface albedo and cloud fraction values associated with different regions accordingly to the Copernicus Climate Change Service (C3S), to asses the local change of the SW DREE due to the increase of pollution in a mineral

dust layer. Figure 8 shows how the SW DREE changes for increasing pollution content into the mineral dust layer for different surface albedos ($R_\mathrm{s} = 0.03, 0.15, 0.28, 0.68$ for ocean, global average, desert, and snow, respectively) in the case of cloud-free conditions $A_\mathrm{c} = 0$. Figure 9 shows how the SW DREE changes for increasing pollution content into the mineral dust layer for different regions with a specific surface albedo and cloud fraction values.

When the mineral dust layer is above a surface with high surface albedo, as a desert or snow, the SW DREE increases

for increasing pollution content. Thus, the cooling effect decreases (or the warming effect increases) for increasing pollution content compared to the case of a pure mineral dust layer. The results of the linear regression analysis between the SW DREE and the pollution contribution in terms of mass $\log_{10}(M_\mathrm{rBC}/M_{\text{super-}\mu\text{m ambient}})$ shows that the relationship between the two variables is statistically significant for cases with high surface albedo ($R_\mathrm{s} \geq 0.23$), as indicated by the $p-$values below the threshold of $0.05$.

When the mineral dust layer is above a surface with low surface albedo, as the ocean, the SW DREE decreases slightly for increasing pollution content. Thus, the cooling effect may increase for increasing pollution compared to the case of a pure mineral dust layer. However, the relationship is not statistically significant, as indicated by the $p-$values above the threshold of $0.05$, and also by the wide confidence interval of the regression parameters. Thus, our data do not provide enough evidence to conclude that the SW DREE changes with increasing pollution content within the mineral dust layer. However, the relationship

may become significant if more pollution is present in the mineral dust layer.

The presence of clouds do not modify the significance of the relationship, it changes the magnitude of the SW DREE and of the slope (Figure 9). Regions close to the Eastern Mediterranean that have high surface albedo are the Saharan desert and the Arabian Peninsula. These regions have a relatively low cloud fraction ($A_\mathrm{c} = 0.21$), which lead to an increase by $+6(\pm 2) W m^{-2} AOD^{-1}$ of the SW DREE per unit increase of $\log_{10}(M_\mathrm{rBC}/M_{\text{super-}\mu\text{m ambient}})$. Turkey has several mountain

chains which surface albedo present a strong annual cycle due to changes in vegetation and snow coverage. We considered in particular the Eastern part of Turkey of the Aras mountains. The annual average surface albedo of $0.23$ and the cloud



fraction of 0.52 leads to absolute numbers of the SW DREE similar to the one of the Saharan desert and the Arabian Peninsula and slope $+3(\pm1)Wm^{-2}AOD^{-1}$. During winter when the mountains are covered by snow the SW DREE increases by $+8(\pm1)Wm^{-2}AOD^{-1}$ per unit increase of $\log_{10}(M_{\text{rBC}}/M_{\text{super-}\mu\text{m ambient}})$. This slope is considerably smaller than the value

of $+24(\pm4)Wm^{-2}AOD^{-1}$ obtained without considering the effect of clouds (Figure 8). Regions with low surface albedo of interest for our measurements are the Mediterranean sea and the territory of Greece. The SW DREE in these regions changes by $-2(\pm2)Wm^{-2}AOD^{-1}$ and $-1(\pm1)Wm^{-2}AOD^{-1}$, respectively, per unit change of $\log_{10}(M_{\text{rBC}}/M_{\text{super-}\mu\text{m ambient}})$.

These results suggest that for regions with a high surface albedo, quantifying the pollution within a mineral dust layer is very important to correctly estimate the local direct radiative effect of the mineral dust layer. Further studies are needed on the

impact of mixing pollution and mineral dust, especially above surfaces with low surface albedo. In addition, the limitations and uncertainties of the SW DREE relationships do not allow for further considerations. The generalization of these results requires calculations with a more sophisticated radiative transfer model, which is beyond the scope of this study..

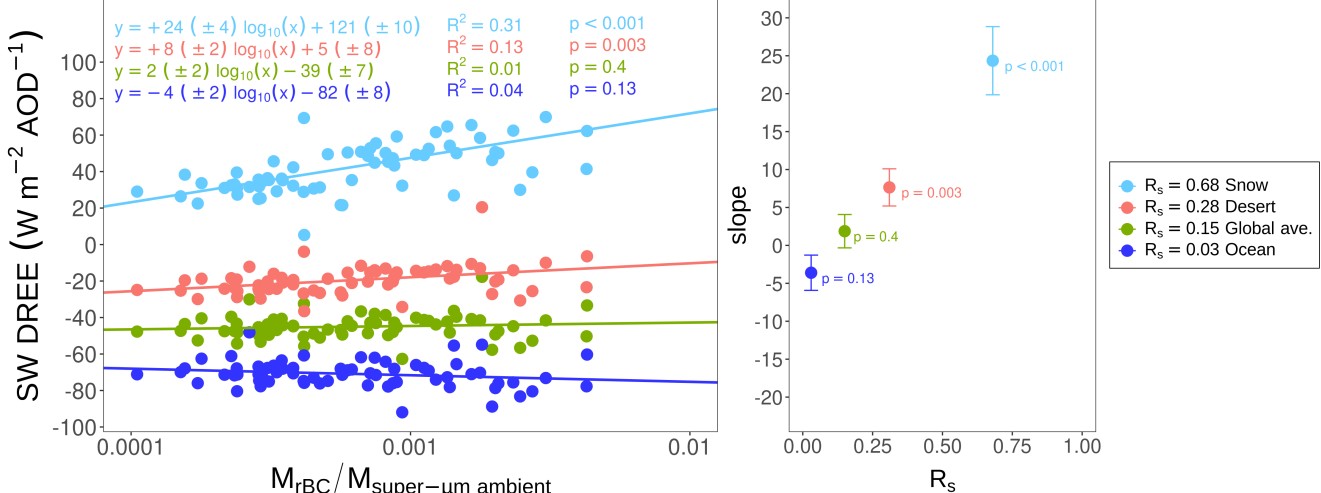

**Figure 8. Change in the short-wave Direct Radiative Effect Efficiency (SW DREE) for increasing pollution in the mineral dust layer for different surface albedos in cloud-free conditions.** The left panel shows the short-wave Direct Radiative effect Efficiency (SW DREE), calculated using a simplified formula (Haywood and Shine, 1995), the measured values of $g$ and SSA at $\lambda = 525\,\text{nm}$ and global average parameters as in Sherman et al. (2015), as a function of the pollution contribution in terms of mass ($M_{\text{rBC}}/M_{\text{super-}\mu\text{m ambient}}$). $M_{\text{super-}\mu\text{m ambient}}$ is the mass of the ambient super$-\mu\text{m}$ particles, since the values of $g$ and SSA used to calculate the SW DREE were extended to the ambient particle size distribution (see section S3 in the supplementary information). Different colors indicate the SW DREE calculated for different surface albedos ($R_s$). The calculations are done for cloud-free conditions ($A_c = 0$). Least squares regression results, including regression line and equation, determination coefficient and $p-$value, are shown in the corresponding color. The right panel shows the slope obtained for each linear regression as function of the surface albedo.



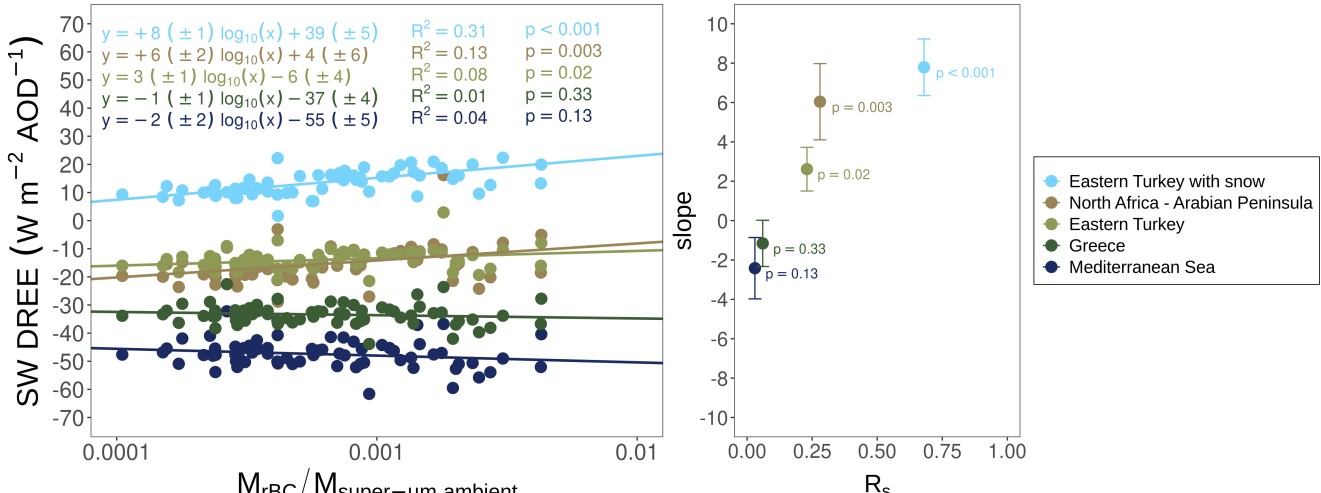

**Figure 9. Change in the short-wave Direct Radiative Effect Efficiency (SW DREE) for increasing pollution in the mineral dust layer for different regions.** The left panel shows the short-wave Direct Radiative Effect Efficiency (SW DREE), calculated using a simplified formula (Haywood and Shine, 1995), the measured values of $g$ and SSA at $\lambda = 525\,\text{nm}$ and global average parameters as in Sherman et al. (2015), as a function of the pollution contribution in terms of mass ($M_{\text{rBC}}/M_{\text{super-}\mu\text{m ambient}}$). $M_{\text{super-}\mu\text{m ambient}}$ is the mass of the ambient super$-\mu$m particles, since the values of $g$ and SSA used to calculate the SW DREE were extended to the ambient particle size distribution (see section S3 in the supplementary information). Different colors indicate the SW DREE calculated for different regions by considering different surface albedo ($R_{\text{s}}$) and cloud fraction ($A_{\text{c}}$) combinations. Annual average $R_{\text{s}}$ and $A_{\text{c}}$ were identified for each region using data from the Copernicus Climate Change Service (C3S): $R_{\text{s}} = 0.03$ and $A_{\text{c}} = 0.33$ for the Mediterranean sea, $R_{\text{s}} = 0.06$ and $A_{\text{c}} = 0.48$ for Greece, $R_{\text{s}} = 0.28$ and $A_{\text{c}} = 0.21$ for Saharan desert and Arabian Peninsula, $R_{\text{s}} = 0.23$ and $A_{\text{c}} = 0.52$ for Eastern Turkey and $R_{\text{s}} = 0.68$ and $A_{\text{c}} = 0.68$ for Eastern Turkey when covered by snow. Least squares regression results, including regression line and equation, determination coefficient and $p-$value, are shown in the corresponding color. The right panel shows the slope obtained for each linear regression as function of the surface albedo.



# 5 Conclusions

During the A-LIFE aircraft field experiment in April 2017, we measured the properties of mineral dust and mineral dust-
pollution mixtures in the Eastern Mediterranean. In this study, we analyzed the intensive aerosol optical properties directly
measured during A-LIFE with an Aurora 4000 polar nephelometer and a Tricolor Absorption Photometer, such as the scatter-
ing, absorption and single scattering albedo Ångström exponent (SAE, AAE, and SSAAE), and the asymmetry parameter and
the single scattering albedo ($g$ and SSA). We investigated how these intensive aerosol optical properties change when mineral
dust layers are mixed with anthropogenic aerosols/pollution in different quantities. In addition, we compared these results for
mineral dust from the two source regions that affect the Eastern Mediterranean: the Arabian peninsula and the Saharan desert.

Our data show that all intensive aerosol optical properties (SAE, AAE, SSAAE, $g$, and SSA) change significantly with
increasing pollution in the mineral dust layer, while the differences between the optical properties of mineral dust from the two
different source regions are not significant. The values of the intensive aerosol optical properties of Saharan and Arabian dust
are similar (within error bars) when comparing data with similar pollution content.

These findings have different implications (i) for the retrieval of mineral dust properties, (ii) for the identification of mineral
dust episodes in the atmosphere, and (iii) for the direct radiative effect of mineral dust layers when mixed with pollution.

We highlight that care is needed when retrieving the optical properties of mineral dust because mineral dust layers often
contain mixtures of other aerosol components. In particular, when analyzing the differences between mineral dust from different
source regions, evaluating the pollution content and comparing only measurements with similar pollution content is essential.

The presence of pollution masks the mineral dust signal because the Ångström exponents (SAE, AAE, and SSAAE) are
significantly affected by the presence of pollution. As a result, the identification of mineral dust cases using the Ångström expo-
nents fails in polluted mixtures. Once the presence of mineral dust has been established by independent methods, the Ångström
exponents may be helpful in estimating the amount of pollution mixed into the mineral dust layer. Thus, the Ångström expo-
nents can be used to select cases where non-dust aerosols have a minimal effect on the optical properties of mineral dust.
Among the Ångström exponents (SAE, AAE, and SSAAE), SAE is the best parameter for estimating the amount of pollution
mixed into a mineral dust layer.

Climate-relevant intensive aerosol optical properties ($g$ and SSA) measured in pure mineral dust layers are in agreement
with values reported by previous studies. However, differences in the values measured in the Falcon aircraft cabin and values
extended to the ambient particle number size distribution point out that differences in the observed intensive aerosol optical
properties in various studies may also be attributed to the discrepancies in the measured particle diameter size range.

The pollution in a mineral dust layer significantly impacts the climate-relevant intensive aerosol optical properties. The
asymmetry parameter and the single scattering albedo ($g$ and SSA) decrease significantly for increasing pollution content,
(e.g., from pure mineral dust to polluted mineral dust median measured values decreased from 0.67 to 0.56 and from 0.96 to
0.89, respectively, at 525 nm). We would like to remind that an increase in pollution indicates not only an increase of black
carbon particles, but also an increase of other fine mode anthropogenic aerosol particles, although we quantified the pollution
using the refractory black carbon mass. Thus, the changes in $g$ and SSA have two opposite impacts on the short-wave direct





radiative effect of the mineral dust layer. The decrease of $g$, due to a decrease in the particle size, leads to an increase in the cooling effect. The decrease of SSA, due to increased absorption, leads to a decrease in the cooling effect (or increase in the warming effect).

Based on a simple calculation of the short-wave direct radiative effect efficiency (SW DREE), these two effects are expected to counteract each other on a global average. However, the SW DREE can have a significant change for increasing pollution content if the calculation is done considering different surface albedos. For instance, if the mineral dust layer is above a surface with high surface albedo, such as desert or snow, the increase in pollution leads to a reduction of the cooling effect (or an increase of the warming effect). On the contrary, if the mineral dust layer is above a surface with low surface albedo, such as

the ocean, an increase in pollution may result in an increase of the cooling effect. Thus, in this case, the effect of decreasing particle size is more significant than the effect of increased absorption. However, it is important to note that this latter effect was not statistically significant in our data.

    Further studies on the impact of mixing pollution and mineral dust are required to generalize these results. A more sophisticated radiative transfer model may be used to assess the impact of mixing pollution and mineral dust on the direct radiative

effect efficiency. In conclusion, an accurate quantification of the pollution mixed in mineral dust layers is necessary because the presence of pollution has major consequences in the identification of mineral dust episodes and the retrieval of mineral dust optical properties and its direct radiative effect.





## Appendix A: Summary of aerosol properties used in this study

**Table A1.** Summary of optical aerosol properties used for the analysis of this study.

| Symbol | Quantity | Size range | Method |
|---|---|---|---|
| $\sigma_{\mathrm{sp},\lambda}^{\mathrm{Aurora\,4000},\alpha}$ | Particle scattering coefficient | in-cabin* | Measured by the Ecotech Aurora 4000 polar nephelometer at three wavelengths ($\lambda = 450, 525, 635$ nm) and for four angular sectors ($\alpha - 180°$, with $\alpha = 0, 20, 50, 90°$). |
| $\sigma_{\mathrm{sp},\lambda}$ | Total particle scattering coefficient | in-cabin* | Measured by the Ecotech Aurora 4000 polar nephelometer at three wavelengths ($\lambda = 450, 525, 635$ nm), for the angular sector $0 - 180°$, and corrected for angular truncation and illumination as indicated in Teri et al. (2022). |
| $\sigma_{\mathrm{ap},\lambda}$ | Particle absorption coefficient | in-cabin* | Measured by the Tricolor Absorption Photometer (TAP, Brechtel Manufacturing Inc., Hayward, USA) at three wavelengths ($\lambda = 465, 520, 640$ nm). |
| $\mathrm{SAE}_{\lambda_{\mathrm{B}}/\lambda_{\mathrm{R}}}$ | Scattering Ångström exponent | in-cabin* | $\mathrm{SAE}_{\lambda_{\mathrm{B}}/\lambda_{\mathrm{R}}} = -log(\sigma_{\mathrm{sp},\lambda_{\mathrm{B}}}/\sigma_{\mathrm{sp},\lambda_{\mathrm{R}}})/log(\lambda_{\mathrm{B}}/\lambda_{\mathrm{R}})$, with the blue and red wavelengths of the Aurora 4000 polar nephelometer ($\lambda_{\mathrm{B}} = 450$ nm and $\lambda_{\mathrm{R}} = 635$ nm). |
| $\mathrm{AAE}_{\lambda_{\mathrm{B}}/\lambda_{\mathrm{R}}}$ | Absorption Ångström exponent | in-cabin* | $\mathrm{AAE}_{\lambda_{\mathrm{B}}/\lambda_{\mathrm{R}}} = -log(\sigma_{\mathrm{ap},\lambda_{\mathrm{B}}}/\sigma_{\mathrm{ap},\lambda_{\mathrm{R}}})/log(\lambda_{\mathrm{B}}/\lambda_{\mathrm{R}})$, with the blue and red wavelengths of the TAP ($\lambda_B = 465$ nm and $\lambda_R = 640$ nm). |
| $\mathrm{SSAAE}_{\lambda_{\mathrm{B}}/\lambda_{\mathrm{R}}}$ | Single scattering albedo Ångström exponent | in-cabin* | $\mathrm{SSAAE}_{\lambda_{\mathrm{B}}/\lambda_{\mathrm{R}}} = -log(\mathrm{SSA}_{\lambda_{\mathrm{B}}}/\mathrm{SSA}_{\lambda_{\mathrm{R}}})/log(\lambda_{\mathrm{B}}/\lambda_{\mathrm{R}})$, where SSA is the single scattering albedo. |
| $g_\lambda$ | Asymmetry parameter | in-cabin* | Calculated using the Müller et al. (2012) approximation using $\sigma_{\mathrm{sp},\lambda}^{\mathrm{Aurora\,4000},\,\alpha}$. |
| $\mathrm{SSA}_\lambda$ | Single scattering albedo | in-cabin* | $\mathrm{SSA}_\lambda = \sigma_{\mathrm{sp},\lambda}/(\sigma_{\mathrm{sp},\lambda} + \sigma_{\mathrm{ap},\lambda})$, where $\lambda$ is the blue, green, or red wavelength of the Aurora 4000 polar nephelometer and of the TAP for $\sigma_{\mathrm{sp},\lambda}$ and $\sigma_{\mathrm{ap},\lambda}$, respectively. |
| $g_{\lambda,\mathrm{ambient}}$ | Asymmetry parameter of the ambient NSD | $10\,\mathrm{nm} - 50\,\mathrm{\mu m}$ | Obtained by extending to the ambient NSD the value of $g_\lambda$ measured in the Falcon aircraft cabin (see section S3 in the supplementary information). |
| $\mathrm{SSA}_{\lambda,\mathrm{ambient}}$ | Single scattering albedo of the ambient NSD | $10\,\mathrm{nm} - 50\,\mathrm{\mu m}$ | Obtained by extending to the ambient NSD the value of SSA measured in the Falcon aircraft cabin (see section S3 in the supplementary information). |
| SW DREE | Short-wave direct radiative effect efficiency | $10\,\mathrm{nm} - 50\,\mathrm{\mu m}$ | Calculated using a simplified formula derived by Haywood and Shine (1995): SW DREE = SW DRE/AOD = $= -DS_0T_{\mathrm{atm}}(1 - A_c)\beta\mathrm{SSA}[(1 - R_{\mathrm{s}})^2 - (2R_{\mathrm{s}}/\mathrm{SSA}\beta)(1 - \mathrm{SSA})]$ where $D = 0.50$ is the fractional day length, $S_0 = 1370 W m^{-2}$ is the solar constant, $T_{\mathrm{atm}} = 0.76$ is the atmospheric transmission, $A_{\mathrm{c}} = 0.60$ is the cloud fraction, $R_{\mathrm{s}} = 0.15$ is the spectrally averaged surface albedo, SSA is the single scattering albedo, and $\beta$ is the average upscatter fraction, ad can be obtained from $g$ via a parametrization from (Wiscombe and Grams, 1976). SSA and $g$ are the measured values at 525 nm of the ambient NSD. |

*aerosol particles passing through the Falcon sampling system, which has a cut-off diameter (i.e., the diameter at which 50% of the particles pass the inlet and the sampling system) between $2.4\,\mathrm{\mu m}$ and $6.4\,\mathrm{\mu m}$ for flight altitudes between 2 and 8 km depending on the particle's density (see Table 2 and Figure 6 in Schöberl et al., 2023).



**Table A2.** Summary of microphysical aerosol properties used for the analysis of this study.

| Symbol | Quantity | Size range | Method |
|---|---|---|---|
| $M_{\mathrm{rBC}}$ | Mass concentration of refractory black carbon | $70 - 980\,\mathrm{nm}$ | Measured by a Single Particle Soot Photometer (SP2, DMT). |
| $N_{\mathrm{coarse}}$ | Number concentration of ambient coarse mode aerosol | $0.5 - 50\,\mu\mathrm{m}$ | Measured by a optical particle counter, the wing-mounted Cloud and Aerosol Spectrometer (CAS) which is part of University of Vienna's wing-mounted second-generation Cloud, Aerosol, and Precipitation spectrometer (UNIVIE CAPS, DMT) |
| Modeled aerosol mass concentration and composition | Modeled mass concentration of individual aerosol types along the flight path | $100\,\mathrm{nm} - 10\,\mu\mathrm{m}$ | Obtained by combining source-receptor sensitivities calculated with the Lagrangian Particle dispersion model FLEXPART (Stohl et al., 1998; Seibert and Frank, 2004; Pisso et al., 2019) with emission data from the Copernicus Atmospheric Monitoring Service (CAMS; CAMS, 2019; Morcrette et al., 2009). For details, see section 2.3.4. |
| In-cabin NSD | Particle number size distribution passing through the Falcon aerosol inlet and transport system | in-cabin* | Obtained by combining measurements of the in-cabin condensation particle counter (CPC, model TSI 3760a, TSI Inc.; in this study referred to as as TSI CPC), the wing-mounted active sampling Ultra-High Sensitivity Aerosol Spectrometer-Airborne (UHSAS-A, DMT), and the in-cabin optical particle counter (SkyOPC, model 1.129, GRIMM Aerosol Technik), and parameterized with a combination of three log-normal distributions. Refractive indices used for the derivation of geometrical particle size from OPC measurements are chosen accordingly to the modeled aerosol composition. |
| Ambient NSD | Ambient particle number size distribution | $10\,\mathrm{nm} - 50\,\mu\mathrm{m}$ | Obtained by combining measurements of the in-cabin condensation particle counter (TSI CPC), the wing-mounted UHSAS-A, the in-cabin SkyOPC, and the wing-mounted CAS component of the Vienna CAPS, and parameterized with a combination of three log-normal distributions. Refractive indices are chosen accordingly to the modeled aerosol composition. |
| $M_{\mathrm{sub\text{-}\mu m}}$ | Mass concentration of sub$-\mu$m particles | $100\,\mathrm{nm} - 1\,\mu\mathrm{m}$ | Calculated from the the in-cabin NSD, assuming particles with a spherical shape and density based on the modeled aerosol composition. |
| $M_{\mathrm{super\text{-}\mu m\ in\text{-}cabin}}$ | Mass concentration of super$-\mu$m particles passing through the Falcon aerosol inlet and transport system | $d > 1\,\mu\mathrm{m}$ in-cabin* | Calculated from the in-cabin NSD, assuming particles with a spherical shape and density based on the modeled aerosol composition. |
| $M_{\mathrm{super\text{-}\mu m\ ambient}}$ | Mass concentration of super$-\mu$m particles | $1 - 50\,\mu\mathrm{m}$ | Calculated from the ambient NSD, assuming particles with a spherical shape and density based on the modeled aerosol composition. |

*aerosol particles passing through the Falcon sampling system, which has a cut-off diameter (i.e., the diameter at which 50% of the particles pass the inlet and the sampling system) between $2.4\,\mu\mathrm{m}$ and $6.4\,\mu\mathrm{m}$ for flight altitudes between 2 and 8 km depending on the particle's density (see Table 2 and Figure 6 in Schöberl et al., 2023).




**Appendix B: Asymmetry parameter and single scattering albedo representative for the ambient NSD**

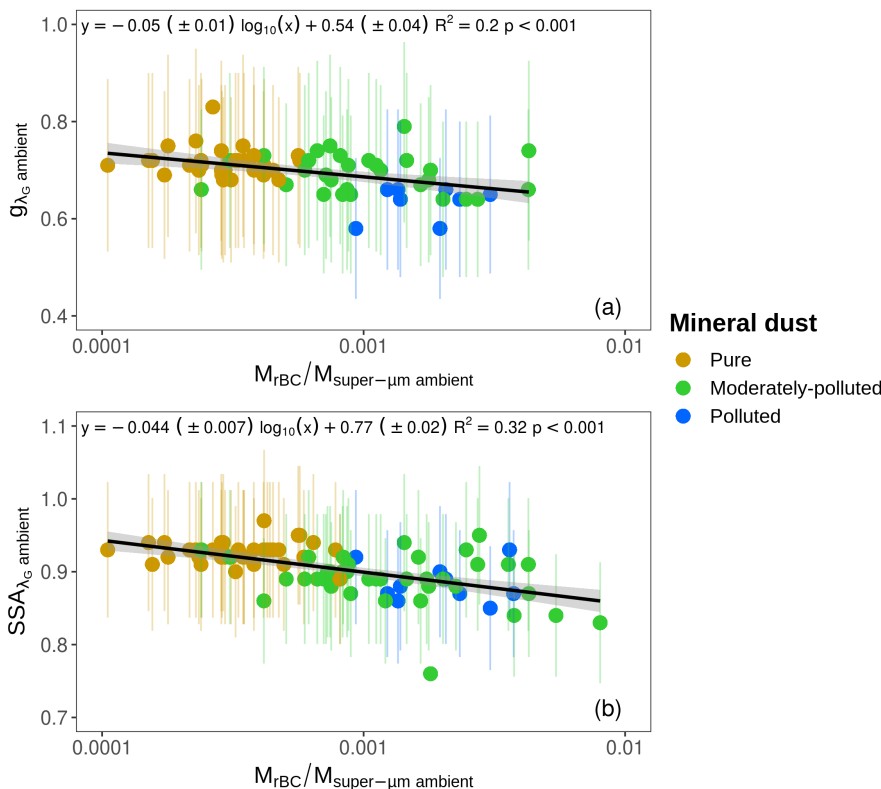

**Figure B1. Intensive aerosol optical properties ($g$, and SSA) representative for the ambient NSD.** Each panel shows a different intensive optical property as a function of the pollution contribution in terms of mass ($M_{rBC}/M_{super\text{-}\mu m\ ambient}$). Panel (a) shows the asymmetry parameter ($g$) at $\lambda = 525$ nm, and panel (b) shows the single scattering albedo (SSA) at $\lambda = 525$ nm Least squares regression results, including regression line and equation, determination coefficient and $p$-value, are shown in black. Color and shape codes indicate the aerosol types following the A-LIFE aerosol classification scheme.



**Table B1.** Summary statistics for each aerosol type of of $g_{\lambda,\text{ambient}}$ and $\text{SSA}_{\lambda,\text{ambient}}$ at three wavelengths ($\lambda_B = 450$ nm, $\lambda_G = 525$ nm, $\lambda_B = 635$ nm) representative for the ambient NSD measured out-cabin of the Falcon aircraft during A-LIFE for sequences classified as mineral dust by the A-LIFE classification scheme Weinzierl et al. (in prep.).

| | | $g_{\lambda_B,\text{ambient}}$ | $g_{\lambda_G,\text{ambient}}$ | $g_{\lambda_R,\text{ambient}}$ | $\text{SSA}_{\lambda_B,\text{ambient}}$ | $\text{SSA}_{\lambda_G,\text{ambient}}$ | $\text{SSA}_{\lambda_R,\text{ambient}}$ |
|---|---|---|---|---|---|---|---|
| | median | **0.72** | **0.71** | **0.72** | **0.91** | **0.93** | **0.95** |
| | $3^{rd}$ | 0.67 | 0.68 | 0.69 | 0.89 | 0.90 | 0.92 |
| Pure mineral dust | $25^{th}$ | 0.70 | 0.70 | 0.71 | 0.90 | 0.92 | 0.95 |
| | $75^{th}$ | 0.74 | 0.72 | 0.76 | 0.92 | 0.93 | 0.96 |
| | $97^{th}$ | 0.79 | 0.77 | 0.79 | 0.94 | 0.95 | 0.97 |
| | median | **0.68** | **0.70** | **0.71** | **0.88** | **0.89** | **0.92** |
| | $3^{rd}$ | 0.61 | 0.64 | 0.60 | 0.84 | 0.83 | 0.84 |
| Moderately-polluted mineral dust | $25^{th}$ | 0.66 | 0.66 | 0.68 | 0.87 | 0.88 | 0.90 |
| | $75^{th}$ | 0.71 | 0.72 | 0.73 | 0.91 | 0.91 | 0.93 |
| | $97^{th}$ | 0.74 | 0.76 | 0.75 | 0.93 | 0.94 | 0.96 |
| | median | **0.56** | **0.56** | **0.54** | **0.89** | **0.89** | **0.88** |
| | $3^{rd}$ | 0.49 | 0.50 | 0.50 | 0.86 | 0.84 | 0.85 |
| Polluted mineral dust | $25^{th}$ | 0.55 | 0.54 | 0.52 | 0.88 | 0.87 | 0.88 |
| | $75^{th}$ | 0.57 | 0.57 | 0.56 | 0.90 | 0.89 | 0.90 |
| | $97^{th}$ | 0.58 | 0.59 | 0.58 | 0.95 | 0.94 | 0.95 |



*Data availability.* The data used to obtain the results of this manuscript will be made publicly available in the University of Vienna data archive Phaidra (https://doi.org/10.25365/phaidra.454)

*Author contributions.* MT and BW conceptualized the study. MT performed the data analysis, and wrote the manuscript with the support of BW. BW coordinated the A-LIFE project. BW and MT performed the aerosol optical properties measurements during A-LIFE with the support of TM. MT performed the data quality assurance of the aerosol optical properties with the support of JG and BW. JG and MT

calculated the simulated aerosol optical properties. BW, MD and KH performed the aerosol microphysical properties measurements during the A-LIFE field campaign. MD, MS, and JG calculated the combined particle number size distribution. KK and SA performed the single particle chemistry analysis of samples collected during A-LIFE. AT and PS performed the simulations with the Lagrangian particle dispersion model FLEXPART. All co-authors participated in the scientific discussion and reviewed the manuscript.

*Competing interests.* The authors declare that they have no conflict of interest.

*Acknowledgements.* The research leading to these results has received funding from the European Research Council (ERC) under the European Union's Horizon 2020 research and innovation programme (grant agreement no. 640458, A-LIFE). K. Kandler is funded by the Deutsche Forschungsgemeinschaft (DFG, German Research Foundation) – 264912134; 378741973; 416816480. We thank ZAMG (now Geosphere) for providing access to ECMWF forecast data to calculate the trajectories in real time, and to the CAMS User Support team. Copernicus Atmosphere Monitoring Service (CAMS) information, partly modified, was used for this paper; neither the European Commis-

sion nor ECMWF is responsible for any use that may be made of the information. We thank the Vienna Doctoral School in Physics (VDSP) for financial support. We acknowledge the use of imagery from the NASA Worldview application (https://worldview.earthdata.nasa.gov), part of the NASA Earth Observing System Data and Information System (EOSDIS).

*Disclaimer.* The views expressed in this study are those of the authors and do not necessarily represent the views of the CTBTO Preparatory Commission





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
