# Peer review of "Pollution affects Arabian and Saharan dust optical properties in the Eastern Mediterranean"

_EGUsphere, 2024_

## Referee Comment (RC2)

**Review of the manuscript submitted for publication to the Journal of Atmospheric Chemistry and Physics entitled "Pollution affects Arabian and Saharan dust optical properties in the Eastern Mediterranean", by Marilena Teri et al. (ref. egusphere-2024-701)**

*By François Dulac, LSCE, 14 June 2024*

**Recommendation**

Major revision

**Reviewer Comments to Authors**

Synthesis:

This manuscript analyses an extensive set of aircraft observations obtained in the eastern Mediterranean troposphere during the A-LIFE field campaign in April 2017. It focuses on the influence of the presence of pollution particles on soil dust particles long-range transported either form the Saharan or the Arabian desert. The paper is written in good English, well structured, and easy to read. I find that the topic is adapted to the scope of the journal and the proposed approach quite sound. Conclusions bring interesting new knowledge on optical properties of dust from the two main source regions affecting the eastern Mediterranean. I do consider that this study deserves publication. I think, however, that a revision is necessary to bring additional information and analyses. I hope that my remarks and suggestions listed below will be helpful.

Major comments:

1- I consider critical that the A-LIFE generic paper by Weinzierl et al., cited "in preparation" in the present manuscript, is not available to the reader. The corresponding reference given in the bibliography is quite approximative and let think that it might not be available in a near future. It is an important reference for the present manuscript, cited 10 times for various reasons. According to me, such a paper definitely needs to be available to the reader of the Teri et al.'s paper since it provides complementary information on the campaign, meteorological conditions, various dust events, aerosol composition and classification, including the aerosol classification methodology. If this A-LIFE generic paper were not available when submitting the revised Teri et al. manuscript, it would be requested to substantially increase methodological information to make the present paper self-consistent, especially regarding the dust cases classification.

2- The absence of information from microscopy analyses on the internal/external mixing nature of dust with pollution particles is a bit striking. The nature of mixing is an important information to understand how pollution particles affect dust particle properties, and variability in the type of mixing, e.g. between Arabian and Saharan dust, might explain some of the differences. I would expect this aspect on mixing is considered in the paper and that additional relevant information is included based on the microscopic investigations performed (see also comment 6).

3- Optical measurements used to derive the various Angström exponents are based on the blue and red channels (section 2.3.2). Given the wavelength dependence of dust absorption, I wonder whether useful additional information could not be derived from additional Angström exponents using also the green wavelength.

4- Suppl. line 78: I do not agree that the comparison between the simulated and directly measured optical properties is out of scope. Showing that the in-cabin optical properties are correctly reproduced by this approach would provide better confidence in the approach used.

5- Suppl. lines 125-126: you should be able to support this statement based on airmass trajectories and satellite products (e.g., see in particular the North African Sand Storm Survey products for dust source regions at https://nascube.univ-lille1.fr/).

6- Suppl. section S4.1 should be enhanced to consolidate relevant results: please specify the number of respective Saharan and Arabian dust particles analysed, how they were selected, and from how many cases of each type; referring to the particle size distribution of the analysed particles would also be helpful to discard a possible bias between Saharan and Arabian dust subsets; we also expect some details on the respective criteria used for defining the 7 particle types; what about possible variability between cases of a given type? What about internal/external mixing (see also comment 2); Figures S4 and F5 compare Arabian and Saharan dust, combining pure dust and moderately polluted dust cases; do we have the same proportion of those two types of cases for Saharan and Arabian dust; relevant information could be provided in section S4.1.

Minor comments:

7- Lines 261-262: there is some inconsistency since the UHSAS is wing-mounted and not an in-cabin instrument behind the inlet system.

8- Line 271: could you specify on which substrate are particles collected and the type of sampler used?

9- Legend of Figure 2: please specify the respective size of the three sub-samples (N= 37, 40, and 10, I assume, following the text).

10- Lines 315-317: I do not understand your conclusion on aerosol optical properties since we are only dealing here with mass contributions of various aerosol components.

11- Figure 3: please specify the respective size of the three sub-samples (note that it seems less than 37 and 10 points are plotted for pure dust and polluted dust, respectively) and plot correlation lines; in panel (a) at least, I would expect 2 significantly different slopes with respective r values computed separately for pure dust and polluted dust cases.

12- About Figure 2 and 3, can you conclude on the absence of differences between Saharan and Arabian dust?

13- I find that we miss comments on Table 1; in particular, could a variance analysis confirm whether the 3 subsets are significantly different or could be possibly reduced to 2?

14- Table 1 and Table 2: please add the respective number of samples of the 3 subsets.

15- Figure 4: please indicate respective number of samples of the 2 subsets.

16- Line 395-396: what about the case of Saharan dust in the western Mediterranean (Denjean et al., 2016)?

17- Lines 411-412: I would expect a figure to illustrate the statement on respective particle size distributions of Arabian and Saharan dust.

18- Line 447: I find that the conclusive statement is too strong and should be formulated a bit differently; I suggest "as sole indicators to detect the presence of mineral dust in case of mixing with pollution aerosols, because […]".

19- Line 531: better specifying "considered" before "representative"; could you support this hypothesis with a reference?

20- Lines 354 and 542: the term "global" is probably not appropriate here; in line 354, do you mean "regional"?

21- Line 618: I would also expect some information on respective variabilities around those numbers.

22- Suppl. line 119: can we consider that the difference is significant (might you provide respective std. dev.)?

23- -Lines 963-964 and Suppl. lines 201-202: the very incomplete reference to the expected paper by Weinzierl and coauthors (in prep.) must be specified, or omitted and the paper content be augmented accordingly (see General Comment 1).

Editorial comments and suggestions:

-Line 34: after "absorption properties" replace " and" by a comma.

-Line 66: here, I would add Denjean et al. (2016) in the list of references.

-Line 73: I assume "my mask" should read "may mask".

-Line 74: I suggest "place" instead of "emplacement".

-Line 87: remove "decreasing".

-Lines 96-99: I suggest a different formulation in a single sentence, e.g.: "However, optical properties of mineral dust from various source regions may be affected differently by mixing with pollution ([the 3 references reordered chronologically]) and this influence needs to be further investigated".

-Lines 106-107: remove "'s" after "dust" (2 times).

-Line 112: prefer "we use" to "we ued".

-Line 125: add "namely, " before "the Arabian Peninsula".

-Line 126: I would speak of "sub-sections".

-Line 136: rather than "in Cyprus", I suggest to specify "south-west of Cyprus" with latitude and longitude of the airport within brackets.

-Lines 138-139: I suggest "from the nearby deserts, the Arabian Peninsula over the easternmost part of the basin and the Sahara over the central basin".

-Line 139: add "over the central basin" at the end of the sentence.

-Legend of Figure 1: I would add ", last accessed 16 January 2024" at the end of the source reference, and remove NASA-Worldview website from the bibliography (line 842).

-Line 221: "sizes" rather than "size".

-Line 225: "measured" rather than "measure".

-Line 229 and bottom note 1 in page 9, 2$^{nd}$ line: reverse order of the 2 references, for chronology.

-Line 236: specify "aerosol" before "single scattering albedo".

-Line 206: please specify origin of the Dolomite dust used for tests; is there any related specification, commercial reference or publication that could be referred to?

-Line 246: change order of references for chronology.

-Line 275: I suggest "an aerosol classification was determined for all 262 flight sequences. Twelve different".

-Line 280: check citation chronology.

-Lines 281-282: remove the last sentence, which is repeated at the end of the following paragraph.

-Line 295: I suggest "We could investigate" instead of "We investigated".

-Line 301: "in terms of" (plural).

-Line 309: "ratios" rather than "rations".

-Legend of Table 1: I would specify "(median and various percentiles)" after "statistics".

-Legend of Figure 4: I would remove all occurrences of "Panel" or "panel", as well as of "shows" after the panel letters.

-Line 384: it seems to be the 1$^{st}$ citation of Formenti et al. (2011) and should then read "2011a" rather than "2011b"; please check all your citations of Formenti et al. (2011) and possibly shift 2011a and b in the bibliography.

-Line 396: move 2007 before 2011.

-Line 416: "illite" rather than "ilite".

-Line 426-427: unclear conclusion to me; do you mean "While the pollution significantly changes the optical properties of dust particles, we cannot conclude that it has the same effect on Arabian and Saharan dust"?

-Line 433: missing verb, possibly change "and that" to "they indicate that".

-Line 451: you could provide the ranges from Lee et al. (2012).

-Line 467: change the ";" sign between references by a comma and add "and" before the last one.

-Line 482: "Figure 6 shows" instead of "Figures 6 show".

-Line 488 and 494, and legends of Figures 6 and 7: I believe it would be needed to read "2011b" rather than "2011a" (see previous comment on Formenti et al., 2011); in the graphic legend of those 2 figures, you should also specify 2011a or b in the reference to Formenti et al., 2011.

-End of line 490: I would add "in various size ranges indicated in Figure 6".

-Line 511: I suggest "is affected by" rather than "considers the effect of".

-Line 512: I suggest "reflects" rather than "considers".

-Lines 520 and 522: I would specify "the surface cooling effect" instead of "the cooling effect".

-Line 522: I suggest changing "(or the warming effect may be enhanced)" by "and the atmospheric warming may be enhanced,".

-Line 547: what about adding "significantly" to "does not change"?

-Line 551: remove "for" after "take into account".

-Line 571: "does not modify".

-Line 573: "which leads".

-Line 586: remove "for".

-End of the legend of Figures 8 and 9: "as a function".

-Figure 9: for better readability, the y-axis of the left plot would better start at -5 than -11, and inserted notes about p values should be written with bigger characters.

-End of legend of Figure 9: I suggest "Respective least squares regression".

-Line 607: I suggest "in case of mixtures with pollution particles" rather than "in polluted mixtures".

-Line 623: since surface cooling has also been mentioned, better specify here "cooling effect at top of atmosphere".

-Table A1, bottom of left column: change "ad" to "and".

-Table A2, left column: change "accordingly to" to "according to" (2 occurrences).

-Figure B1 and Table B1: report number of points.

-Line 653: trajectories of what were calculated in real time? The link with the present work is not clear, is this mention necessary?

-Line 654: "partly modified" CAMS information appeals some explanation.

-Line 697: remove the sign "*".

-Supplement line 25: remove the spare article "a".

-Suppl. line 31: "which modifies".

-Suppl. line 39: "with the particle"

-Suppl. line 40: move "at one wavelength" after "Mm$^{-1}$"; not clear to me why using the italic style for this unit.

-Suppl. lines 52 and 59: I'd rather use "at" than "for" about wavelengths.

-Suppl. Line 59: "Arctic" (missing "c"); also remove "of" before the bracket.

-Suppl. Line 65: "in-cabin measured".

-Suppl. Lines 65, 66 and 78: I'd rather write "representative of".

-Suppl. Line 76: please add a reference for the assumed refractive index of sulfate.

-Suppl. line 78: add "and" after "NSDs".

-Suppl. line 86: I suggest to specify the impactor type: "A Micro Inertial Impactor (MINI) sampler" as far as I understand from Kandler et al. (2007); this single ref. seems appropriate here, if the 2011 were also cited, put 2007 first; the Schöberl et al. (2023) reference concerns the inlet efficiency, not the impactor; please clarify if it remains cited.

-Suppl. line 90: please specify the microscope model?

-Suppl. line 101: "of results of".

-Suppl. line 108: I suggest to remove " (21%)" and to replace by " (21% against 8% for Arabian dust)" the last part of the sentence starting by ", while".

-Suppl. line 110: also indicate carbonate percentage value for Saharan dust within brackets.

-Suppl. lines 111-112: please rephrase to clarify.

-Suppl. line 115: "dust mineralogical characteristic".

-Suppl. line 116: "compared to Saharan dust,".

-Suppl. line 117: I would cite Figure S3.

-Suppl. line 118: could you briefly summarize the Di Biagio et al.'s method?

-Suppl. line 122: cite Figure S5, which is presently not called in the text.

-Suppl. line 125: can you provide values for the Atlantic region?

-Figure S3 legend: not clear whether the class "Oxides hydroxydes" is only related to iron? The definition of particles classes should help (see comment 5).

-Figure S4 legend: "The graph shows three elemental ratios".

---

## Author Response (AR1)

**Reply to review #1 of the manuscript: „Pollution affects Arabian and Saharan dust optical properties in the Eastern Mediterranean" by Teri et al.**

We would like to thank Darrel Baumgardner for the thoughtful comments which helped to improve the manuscript. In the following, the comments raised by the reviewer are marked **in bold**. Our replies are given in black.

**Referee #1: Darrel Baumgardner**

**Assigning ratings to this manuscript is challenging, as you can see from my "Outstanding" rating for scientific significance, because I think this is worthy of publication, and the "Excellence" for presentation is because I found the organization of the paper to be very logical and I was able to easily follow the flow. The figures, while numerous, are important for highlighting the points that the authors are making throughout the text.**

**The challenge that I refer to is seen in my "good" rating for the scientific quality and in my decision that the manuscript needs a major revision. The scientific quality is actually better than just "good" and it is likely I can be convinced to recommend acceptance once the authors address my primary concern about this study. I selected those particular ratings to get the authors' attention and recognize that I am serious about this concern.**

**I am aware that the primary focus of this study is on the measurement of the intrinsic, optical properties of ensembles of particles that are captured on the TAP filters and that pass through the sample volume of the polar nephelometer. I have no argument whatsoever about the approach taken and the conclusions drawn.**

**What puzzles me is that there is no discussion that I could find on the relative impact on these optical properties on how the pollution, i.e. rBC is mixed with the dust, i.e. are the rBC and dust mixed externally or internally? I don't think that this has a trivial response and it could have an impact on the final conclusions. Is the optical response of the nephelometer and TAP the same, regardless of how the dust and rBC are mixed?**

**The University of Vienna group are world leaders in the analysis of SP2 measurements and yet one of the features of the SP2 has been ignored here, i.e. derivation of the mixing state or effective coating thickness. This metric would have allowed a very quantitative evaluation of what percentage of the dust had no rBC mixed with it and what percentage was mixed. This is not a trivial question and one that has direct bearing on the reported results.**

**Once I get a response from the authors, I will do a reassessment of my ratings.**

We agree that including information on the aerosol mixing state would be very interesting from a scientific perspective. For example, the mixing state can influence the optical properties (e.g., Liu et al. 2015), and the use of a simplified mixing state in model simulations can lead to significant uncertainties in the derived optical properties (Yao et al. 2022).

Since the manuscript is already extensive, we have tried to find a balance between including more material, but without making the paper much longer. Nevertheless, we had a closer look at the SP2 data. We identified rBC particles with a heavy coating through the delay time of the peaks of the incandescent signal with respect to the peak of the scattering signal. The results of the analysis are shown in the new Figure 6.

We believe that not only the mixing state of the rBC particles is important, but the mixing state of all particles including mineral dust in a given measurement period. For this additional analysis, we focussed on 3 cases of airborne impactor samples collected in pure and moderately polluted mineral dust sequences. We used a scanning electron microscope (ESEM) for the analysis of particles with projected area diameter (PAD) in the range 500 nm - 3 µm, and a transmission electron microscope (TEM) for particles with PAD < 500 nm.

Results of this additional analysis are included in the new section 2.3.6 in the manuscript.

The 3 investigated case studies, however, are too limited to assess the response of the nephelometer and the TAP for different mixing states. A detailed investigation would be a study on its own.

**References**

Liu, C., Chung, C. E., Yin, Y., and Schnaiter, M.: The Absorption Ångström Exponent of Black Carbon: From Numerical Aspects, Atmospheric Chemistry and Physics, 18, 6259–6273, https://doi.org/10.5194/acp-18-6259-2018, 2018.

Yao, Y., Curtis, J. H., Ching, J., Zheng, Z., and Riemer, N.: Quantifying the Effects of Mixing State on Aerosol Optical Properties, Atmospheric Chemistry and Physics, 22, 9265–9282, https://doi.org/10.5194/acp-22-9265-2022, 2022.

**Reply to review #2 of the manuscript: „Pollution affects Arabian and Saharan dust optical properties in the Eastern Mediterranean" by Teri et al.**

We would like to thank François Dulac for the thoughtful and detailed comments which helped to improve the manuscript. In the following, the questions and comments raised by the reviewer are marked **in bold**. Our answers are given in black.

**Referee #2: François Dulac**

**This manuscript analyses an extensive set of aircraft observations obtained in the eastern Mediterranean troposphere during the A-LIFE field campaign in April 2017. It focuses on the influence of the presence of pollution particles on soil dust particles long-range transported either form the Saharan or the Arabian desert. The paper is written in good English, well structured, and easy to read. I find that the topic is adapted to the scope of the journal and the proposed approach quite sound. Conclusions bring interesting new knowledge on optical properties of dust from the two main source regions affecting the eastern Mediterranean. I do consider that this study deserves publication. I think, however, that a revision is necessary. I hope that my remarks and suggestions listed in the attached file will be helpful.**

**Major comments:**

**1- I consider critical that the A-LIFE generic paper by Weinzierl et al., cited "in preparation" in the present manuscript, is not available to the reader. The corresponding reference given in the bibliography is quite approximative and let think that it might not be available in a near future. It is an important reference for the present manuscript, cited 10 times for various reasons. According to me, such a paper definitely needs to be available to the reader of the Teri et al.'s paper since it provides complementary information on the campaign, meteorological conditions, various dust events, aerosol composition and classification, including the aerosol classification methodology. If this A-LIFE generic paper were not available when submitting the revised Teri et al. manuscript, it would be requested to substantially increase methodological information to make the present paper self-consistent, especially regarding the dust cases classification.**

We critically reviewed the description of the A-LIFE aerosol classification scheme and extended the section to give more detail about the A-LIFE aerosol classification scheme.

Additionally, we reviewed the citations of the upcoming A-LIFE overview paper and removed citations where possible.

We think that with these modifications, the manuscript is a stand-alone paper and can be published before the A-LIFE overview paper is available.

**2- The absence of information from microscopy analyses on the internal/external mixing nature of dust with pollution particles is a bit striking. The nature of mixing is an important information to understand how pollution particles affect dust particle properties, and variability in the type of mixing, e.g. between Arabian and Saharan dust, might explain some of the differences. I would expect this aspect on mixing is considered in the paper and that additional relevant information is included based on the microscopic investigations performed (see also comment 6).**

We agree with the reviewer that mixing of mineral dust with anthropogenic compounds

influences the particle's optical properties as well as their activation properties as cloud condensation and ice nuclei.

We have tried to find a balance between including more material, but without making the paper much longer.

Following the reviewers request, we additionally analysed 3 airborne impactor samples (<1μm diameter) with the Transmission Electron Microscope (TEM) to obtain information on the composition and possible mixing state in pure and moderately polluted mineral dust sequences.

Results of this additional analysis are included in the new section 2.3.6 in the manuscript.

**3- Optical measurements used to derive the various Angström exponents are based on the blue and red channels (section 2.3.2). Given the wavelength dependence of dust absorption, I wonder whether useful additional information could not be derived from additional Angström exponents using also the green wavelength.**

The Angström exponents of scattering, absorption, and single scattering albedo (i.e., their wavelengths dependencies: SAE, AAE, SSAAE), were obtained not only by using wavelengths pairs of the two extreme wavelengths (blue = 450 nm, red = 635nm), but also by including the blue and green (450 and 525 nm) as well as the green and red (525 and 635nm) wavelength pairs.

Results from both methods are mostly similar suggesting that no additional information is obtained by using the other wavelengths. Therefore, we decided to only include the wavelength pair with the lowest uncertainty, i.e. blue and red.

**4- Suppl. line 78: I do not agree that the comparison between the simulated and directly measured optical properties is out of scope. Showing that the in-cabin optical properties are correctly reproduced by this approach would provide better confidence in the approach used.**

We revised section S3 of the supplementary information by including the comparison between the in-cabin simulated and directly measured intensive optical properties (see Section S3 and Figure S3).

The directly measured intensive optical properties are reproduced by this approach within ~16 % for $g_{\lambda,\text{in-cabin}}$ and within ~10 % for $SSA_{\lambda,\text{in-cabin}}$. This comparison provides evidence that the approach used to extend the in-situ measured optical properties to ambient conditions is sufficiently accurate. A deeper analysis of the differences between model simulations and measured optical properties is out of the scope of this paper.

**5- Suppl. lines 125-126: you should be able to support this statement based on airmass trajectories and satellite products (e.g., see in particular the North African Sand Storm Survey products for dust source regions at https://nascube.univ-lille1.fr/).**

Given the low number of samples, it is not possible to identify which specific process is responsible for the observed differences. Therefore, we have removed the sentences from the paper.

**6- Suppl. section S4.1 should be enhanced to consolidate relevant results: please specify the number of respective Saharan and Arabian dust particles analysed, how they were selected,**

and from how many cases of each type; referring to the particle size distribution of the analysed particles would also be helpful to discard a possible bias between Saharan and Arabian dust subsets; we also expect some details on the respective criteria used for defining the 7 particle types; what about possible variability between cases of a given type? What about internal/external mixing (see also comment 2); Figures S4 and F5 compare Arabian and Saharan dust, combining pure dust and moderately polluted dust cases; do we have the same proportion of those two types of cases for Saharan and Arabian dust; relevant information could be provided in section S4.1.

We have now included the number of particles analysed in the figure and in the text.

During the entire A-LIFE campaign, 77 airborne impactor samples were collected from which 71 were analysed. Each analysed sample was sorted into one of the 12 categories of the A-LIFE aerosol classification scheme. To get a significant number of particles, samples of 'pure' and 'moderately polluted' dust periods were grouped. 6 samples altogether containing ~4000 particles form the Arabian dust subset, and 11 samples (~3000 particles) form the Saharan dust subset.

The criteria for defining the 7 particle types follow the recently publised Panta et al., 2023 methodology. The parametrization can be downloaded from https://doi.org/10.5194/acp-23-3861-2023-supplement.

The manuscript was further modified to give more detail about the mixing state including additional figures and their description.

**Minor comments:**

**7- Lines 261-262: there is some inconsistency since the UHSAS is wing-mounted and not an in-cabin instrument behind the inlet system.**

Thank you for pointing out this inconsistency. The UHSAS-A (nominal size range: 60 nm – 1 µm diameter) is mounted under the wing of the research aircraft, but it covers the size range of particles which also enter the inlet of the Falcon research aircraft (for details see Schöberl et al., 2024). For the calculation of the in-cabin number size distribution (NSD), only UHASA-A data for the size range 125-400 nm were used to complement the measurements of other in-cabin instruments. In the revised manuscript, we modified the definition of the in-cabin NSD to "the particle number size distribution representing the size range passing through the Falcon aerosol inlet and transport system."

**8- Line 271: could you specify on which substrate are particles collected and the type of sampler used?**

We used TEM grids. The details about these measurements are now included in Section S4.

**9- Legend of Figure 2: please specify the respective size of the three sub-samples (N= 37, 40, and 10, I assume, following the text).**

Yes, this is correct. We included the information about the sub-sample size in the figure caption.

**10- Lines 315-317: I do not understand your conclusion on aerosol optical properties since we are only dealing here with mass contributions of various aerosol components.**

We did not intend to draw conclusions about optical properties, but wanted to point out the equivalence of choosing one of the two ratios ($M_{sub-\mu m}/M_{super-\mu m\ in-cabin\ and}$ and $M_{rBC}/M_{super-\mu m\ in-cabin}$) for classifying the pollution contribution in terms of mass inside a mineral dust layer. We modified the paragraph to be more clear.

**11- Figure 3: please specify the respective size of the three sub-samples (note that it seems less than 37 and 10 points are plotted for pure dust and polluted dust, respectively) and plot correlation lines; in panel (a) at least, I would expect 2 significantly different slopes with respective r values computed separately for pure dust and polluted dust cases.**

The respective sizes of the three sub-samples are specified in the figure caption. The number of plotted points are 37, 40, and 10 for pure, moderately-polluted, and polluted mineral dust, respectively. However, the number of data points seems to be lower because some data points overlap. We reduced the point size slightly to make the overlapping data points more visible.

Correlation lines were added to the plots. In panel (a) the correlation lines and respective correlation coefficients r were computed separately for pure, moderately-polluted and polluted mineral dust. The text related to this figure was modified.

**12- About Figure 2 and 3, can you conclude on the absence of differences between Saharan and Arabian dust?**

When we started our analysis, our hypothesis was that there are differences between Saharan and Arabian dust. However, as visible in Figure 7, values of the intensive aerosol optical properties of Saharan and Arabian dust are similar (within error bars) when comparing data with similar pollution content. We discussed to show different plots for Saharan and Arabian dust. However, keeping the length of the manuscript in mind, we decided to leave out the distinction between the dust types. We revised the text to make things more clear.

**13- I find that we miss comments on Table 1; in particular, could a variance analysis confirm whether the 3 subsets are significantly different or could be possibly reduced to 2?**

Results from analysis of variance (ANOVA) and post-hoc test (Tukey's HSD) indicate that the three subsets are significantly different for $SAE_{\lambda B/\lambda R}$, g at all three wavelengths and SSA at the red wavelengths. On the contrary, the difference between moderately-polluted and polluted mineral dust is not statistically significant for $AAE_{\lambda B/\lambda R}$ and SSA at the blue and green wavelengths. Additionally, for $SSAAE_{\lambda B/\lambda R}$, the difference between pure mineral dust and moderately polluted mineral dust is not statistically significant. This result highlights that the sequences classified as moderately-polluted mineral dust present optical properties that overlap either with the polluted or moderately polluted cases. Given the results of the analysis of variance, we think that it is important to keep the three categories: pure, moderately-polluted and polluted dust.

We extended the description of Table 1 and also included the results of the variance analysis in the text.

**14- Table 1 and Table 2: please add the respective number of samples of the 3 subsets.**

Done

**15- Figure 4: please indicate respective number of samples of the 2 subsets.**

Done

**16- Line 395-396: what about the case of Saharan dust in the western Mediterranean (Denjean et al., 2016)?**

We removed the entire paragraph (lines 395 - 398). See also reply to comment 5.

**17- Lines 411-412: I would expect a figure to illustrate the statement on respective particle size distributions of Arabian and Saharan dust.**

The manuscript already is extensive, and we did not want to increase content even more. Therefore, we decided that information on the particle size distribution for Arabian and Saharan dust will be the subject of a separate publication. Thus, the statement was removed from the manuscript.

**18- Line 447: I find that the conclusive statement is too strong and should be formulated a bit differently; I suggest "as sole indicators to detect the presence of mineral dust in case of mixing with pollution aerosols, because […]".**

The conclusive sentence was modified as suggested.

**19- Line 531: better specifying "considered" before "representative"; could you support this hypothesis with a reference?**

Thank you for this suggestion. We modified the text accordingly Other authors have also selected a single wavelength. For example, Haywood and Shine (1995) used 700 nm, while Sherman et al. 2015 used 550 nm. Among the wavelengths at which we had measurements of the optical properties available, we have chosen the green wavelength as representative because it is in the middle of the visible solar spectrum. Results using the blue and red wavelengths were similar.

**20- Lines 354 and 542: the term "global" is probably not appropriate here; in line 354, do you mean "regional"?**

Yes, the DREE is, of course, always "regional". With "global", we meant to indicate that the DREE was calculated using the global average surface albedo (R = 0.15 as in Sherman et al. 2015). We modified the text for clarity.

**21- Line 618: I would also expect some information on respective variabilities around those numbers.**

These numbers come from Table 1, where medians and various percentiles are reported. However, we added values of the 3rd and 97th percentiles in brackets after each number in the text.

**22- Suppl. line 119: can we consider that the difference is significant (might you provide respective std. dev.)?**

For better statistical certainty bootstrapping was done with 10,000 particles for the dust sequences. The result shows a ~10% increase in the total iron oxide weight in Saharan dust (median: 8.65 at a 95% confidence interval of 8.2-8.9) as compared to the Arabian dust median value of 7.8 (95% confidence interval of 7.6-8.1).

We included the statistical parameters in section S4.

**23- Lines 963-964 and Suppl. lines 201-202: the very incomplete reference to the expected paper by Weinzierl and coauthors (in prep.) must be specified, or omitted and the paper content be augmented accordingly (see General Comment 1).**

This paper is in preparation, and it is also cited in Schöberl et al. 2024. The citation was corrected consistently.

Weinzierl B., Dollner M., Gasteiger J., Teri M., Schöberl M., Heimerl K., Kupc A., Tipka A., Seibert P., Huntrieser H., Wagner R., Kandler K., Sudharaj A., Müller T., Brilke S., Fölker N., Sauer D., Reitebuch O., Groß S., Freudenthaler V., Toledano C., Haarig M., Mamouri R., Amiridis V., Ansmann A, and the A-LIFE Science Team: Investigating mineral dust mixtures in the Eastern Mediterranean: results from the A-LIFE aircraft field experiment, in preparation, 2024.

**Editorial comments and suggestions:**

**-Line 34: after "absorption properties" replace " and" by a comma.**

Done

**-Line 66: here, I would add Denjean et al. (2016) in the list of references.**

Done

**-Line 73: I assume "my mask" should read "may mask".**

Done. Thank you for spotting this typo.

**-Line 74: I suggest "place" instead of "emplacement".**

Done

**-Line 87: remove "decreasing".**

Done

**-Lines 96-99: I suggest a different formulation in a single sentence, e.g.: "However, optical properties of mineral dust from various source regions may be affected differently by mixing with pollution ([the 3 references reordered chronologically]) and this influence needs to be further investigated".**

Thank you for the suggestion which we included.

**-Lines 106-107: remove "'s" after "dust" (2 times).**

Done

**-Line 112: prefer "we use" to "we used".**

Done.

**-Line 125: add "namely, " before "the Arabian Peninsula".**

Done

**-Line 126: I would speak of "sub-sections".**

Done

**-Line 136: rather than "in Cyprus", I suggest to specify "south-west of Cyprus" with latitude and longitude of the airport within brackets.**

Done

**-Lines 138-139: I suggest "from the nearby deserts, the Arabian Peninsula over the easternmost part of the basin and the Sahara over the central basin".**

Done

**-Line 139: add "over the central basin" at the end of the sentence.**

Done

**-Legend of Figure 1: I would add ", last accessed 16 January 2024" at the end of the source reference, and remove NASA-Worldview website from the bibliography (line 842).**

Done

**-Line 221: "sizes" rather than "size".**

Done

**-Line 225: "measured" rather than "measure".**

Done

**-Line 229 and bottom note 1 in page 9, 2nd line: reverse order of the 2 references, for chronology.**

Done

**-Line 236: specify "aerosol" before "single scattering albedo".**

Done: we used „particle"

**-Line 206: please specify origin of the Dolomite dust used for tests; is there any related specification, commercial reference or publication that could be referred to?**

We are not sure what the reviewer is referring to here. In line 206, the single scattering albedo Angström exponent is defined.

**-Line 246: change order of references for chronology.**

Done

**-Line 275: I suggest "an aerosol classification was determined for all 262 flight sequences. Twelve different".**

Done

**-Line 280: check citation chronology.**

Done

**-Lines 281-282: remove the last sentence, which is repeated at the end of the following paragraph.**

Done

**-Line 295: I suggest "We could investigate" instead of "We investigated".**

Done

**-Line 301: "in terms of" (plural).**

Done

**-Line 309: "ratios" rather than "rations".**

Done

**-Legend of Table 1: I would specify "(median and various percentiles)" after "statistics".**

Done

**-Legend of Figure 4: I would remove all occurrences of "Panel" or "panel", as well as of "shows" after the panel letters.**

Done

**-Line 384: it seems to be the 1st citation of Formenti et al. (2011) and should then read "2011a" rather than "2011b"; please check all your citations of Formenti et al. (2011) and possibly shift 2011a and b in the bibliography.**

There are two Formenti et al. 2011 papers cited (Formenti et al., 2011a in line 66, and Formenti et al., 2011b in line 384):

Formenti, P., Rajot, J. L., Desboeufs, K., Saïd, F., Grand, N., Chevaillier, S., and Schmechtig, C.: Airborne Observations of Mineral Dust over Western Africa in the Summer Monsoon Season: Spatial and Vertical Variability of Physico-Chemical and Optical Properties, Atmospheric Chemistry and Physics, 11, 6387–6410, https://doi.org/10.5194/acp-11-6387-2011, 2011a.

Formenti, P., Schütz, L., Balkanski, Y., Desboeufs, K., Ebert, M., Kandler, K., Petzold, A., Scheuvens, D., Weinbruch, S., and Zhang, D.: Recent Progress in Understanding Physical and Chemical Properties of African and Asian Mineral Dust, Atmospheric Chemistry and Physics, 11, 8231–8256, https://doi.org/10.5194/acp-11-8231-2011, 2011b.

**-Line 396: move 2007 before 2011.**

Done

**-Line 416: "illite" rather than "ilite".**

Done

**-Line 426-427: unclear conclusion to me; do you mean "While the pollution significantly changes the optical properties of dust particles, we cannot conclude that it has the same effect on Arabian and Saharan dust"?**

Thank you for pointing this out. We reformulated the corresponding sentence to make it more clear.

**-Line 433: missing verb, possibly change "and that" to "they indicate that".**

Done

**-Line 451: you could provide the ranges from Lee et al. (2012).**

Ranges from Lee et al.2012 were included.

**-Line 467: change the ";" sign between references by a comma and add "and" before the last one.**

Done

**-Line 482: "Figure 6 shows" instead of "Figures 6 show".**

Done

**-Line 488 and 494, and legends of Figures 6 and 7: I believe it would be needed to read "2011b" rather than "2011a" (see previous comment on Formenti et al., 2011); in the graphic legend of those 2 figures, you should also specify 2011a or b in the reference to Formenti et al., 2011.**

Thank you for spotting this inconsistency. The label in the figure legend was corrected.

**-End of line 490: I would add "in various size ranges indicated in Figure 6".**

Done

**-Line 511: I suggest "is affected by" rather than "considers the effect of".**

Done

**-Line 512: I suggest "reflects" rather than "considers".**

Done

**-Lines 520 and 522: I would specify "the surface cooling effect" instead of "the cooling effect".**

Done

**-Line 522: I suggest changing "(or the warming effect may be enhanced)" by "and the atmospheric warming may be enhanced,".**

Done

**-Line 547: what about adding "significantly" to "does not change"?**

Done

**-Line 551: remove "for" after "take into account".**

Done

**-Line 571: "does not modify".**

Done

**-Line 573: "which leads".**

Done

**-Line 586: remove "for".**

Done

**-End of the legend of Figures 8 and 9: "as a function".**

Done

**-Figure 9: for better readability, the y-axis of the left plot would better start at -5 than -11, and inserted notes about p values should be written with bigger characters.**

Figures 8 and 9 were modified as recommended.

**-End of legend of Figure 9: I suggest "Respective least squares regression".**

Done

**-Line 607: I suggest "in case of mixtures with pollution particles" rather than "in polluted mixtures".**

Done

**-Line 623: since surface cooling has also been mentioned, better specify here "cooling effect at top of atmosphere".**

Done

**-Table A1, bottom of left column: change "ad" to "and".**

Done

**-Table A2, left column: change "accordingly to" to "according to" (2 occurrences).**

Done

**-Figure B1 and Table B1: report number of points.**

Done

**-Line 653: trajectories of what were calculated in real time? The link with the present work is not clear, is this mention necessary?**

Thank you for pointing this out. It is not necessary to mention this. Therefore, it was removed.

**-Line 654: "partly modified" CAMS information appeals some explanation.**

"Partly modified" was removed, and the formulation of the acknowledgement was modified.

**-Line 697: remove the sign "*".**

Done

**-Supplement line 25: remove the spare article "a".**

Done

**-Suppl. line 31: "which modifies".**

Done

**-Suppl. line 39: "with the particle"**

Done

**-Suppl. line 40: move "at one wavelength" after "Mm-1"; not clear to me why using the italic style for this unit.**

Done.

**-Suppl. lines 52 and 59: I'd rather use "at" than "for" about wavelengths.**

Done

**-Suppl. Line 59: "Arctic" (missing "c"); also remove "of" before the bracket.**

Done

**-Suppl. Line 65: "in-cabin measured".**

In the revised manuscript, we modified the definition of the in-cabin NSD to "the particle number size distribution representing the size range passing through the Falcon aerosol inlet and transport system." See reply to comment 7.

**-Suppl. Lines 65, 66 and 78: I'd rather write "representative of".**

Done

**-Suppl. Line 76: please add a reference for the assumed refractive index of sulfate.**

The reference to Toon et al. 1976 was added.

**-Suppl. line 78: add "and" after "NSDs".**

Done

**-Suppl. line 86: I suggest to specify the impactor type: "A Micro Inertial Impactor (MINI) sampler" as far as I understand from Kandler et al. (2007); this single ref. seems appropriate here, if the 2011 were also cited, put 2007 first; the Schöberl et al. (2023) reference concerns the inlet efficiency, not the impactor; please clarify if it remains cited.**

Done.

**-Suppl. line 90: please specify the microscope model?**

We used a scanning electron microscope (FEI ESEM Quanta 400 FEG instrument, Eindhoven, The Netherlands ) coupled with an X-Max 150  energy-dispersive X-ray spectroscopy (EDX) silicon drift X-ray detector (Oxford). More information about the chemical composition measurements was added in section 2.2, and in section S4.

**-Suppl. line 101: "of results of".**

Done

**-Suppl. line 108: I suggest to remove " (21%)" and to replace by " (21% against 8% for Arabian dust)" the last part of the sentence starting by ", while".**

Done

**-Suppl. line 110: also indicate carbonate percentage value for Saharan dust within brackets.**

The percentage value for Saharan dust (6%)  was included.

**-Suppl. lines 111-112: please rephrase to clarify.**

The sentence was reformulated for clarification.

**-Suppl. line 115: "dust mineralogical characteristic".**

Done

**-Suppl. line 116: "compared to Saharan dust,".**

Done

**-Suppl. line 117: I would cite Figure S3.**

Done

**-Suppl. line 118: could you briefly summarize the Di Biagio et al.'s method?**

We reformulated the sentence, and included more information on the method.

**-Suppl. line 122: cite Figure S5, which is presently not called in the text.**

Done

**-Suppl. line 125: can you provide values for the Atlantic region?**

The value 0.001-0.004 at 520 nm Cape Verde (Haywood et al., 2008) was included.

**-Figure S3 legend: not clear whether the class "Oxides hydroxydes" is only related to iron? The definition of particles classes should help (see comment 5).**

This class contains hematite-like and anatase-like particles with Fe and Ti X-ray signals. In this study, we are only focusing on Fe-oxyhydroxides.

Classification details are already published as open-access data in Panta et al., 2023.

**-Figure S4 legend: "The graph shows three elemental ratios".**

Done

**References:**

Formenti, P., Rajot, J. L., Desboeufs, K., Saïd, F., Grand, N., Chevaillier, S., and Schmechtig, C.: Airborne Observations of Mineral Dust over Western Africa in the Summer Monsoon Season: Spatial and Vertical Variability of Physico-Chemical and Optical Properties, Atmospheric Chemistry and Physics, 11, 6387–6410, https://doi.org/10.5194/acp-11-6387-2011, 2011a.

Formenti, P., Schütz, L., Balkanski, Y., Desboeufs, K., Ebert, M., Kandler, K., Petzold, A., Scheuvens, D., Weinbruch, S., and Zhang, D.: Recent Progress in Understanding Physical and Chemical Properties of African and Asian Mineral Dust, Atmospheric Chemistry and Physics, 11, 8231–8256, https://doi.org/10.5194/acp-11-8231-2011, 2011b.

Panta, A., Kandler, K., Alastuey, A., González-Flórez, C., González-Romero, A., Klose, M., Querol, X., Reche, C., Yus-Díez, J., and Pérez García-Pando, C.: Insights into the single-particle composition, size, mixing state, and aspect ratio of freshly emitted mineral dust from field measurements in the Moroccan Sahara using electron microscopy, Atmos. Chem. Phys., 23,

3861–3885, https://doi.org/10.5194/acp-23-3861-2023, 2023.

Schöberl, M., Dollner, M., Gasteiger, J., Seibert, P., Tipka, A., and Weinzierl, B.: Characterization of the airborne aerosol inlet and transport system used during the A-LIFE aircraft field experiment, Atmos. Meas. Tech., 17, 2761–2776, https://doi.org/10.5194/amt-17-2761-2024, 2024.

Weinzierl B., Dollner M., Gasteiger J., Teri M., Schöberl M., Heimerl K., Kupc A., Tipka A., Seibert P., Huntrieser H., Wagner R., Kandler K., Sudharaj A., Müller T., Brilke S., Fölker N., Sauer D., Reitebuch O., Groß S., Freudenthaler V., Toledano C., Haarig M., Mamouri R., Amiridis V., Ansmann A, and the A-LIFE Science Team: Investigating mineral dust mixtures in the Eastern Mediterranean: results from the A-LIFE aircraft field experiment, in preparation, 2024.

---

## Referee Report (RR1)

**Review of the revised manuscript submitted for publication to the Journal of Atmospheric Chemistry and Physics entitled "Pollution affects Arabian and Saharan dust optical properties in the Eastern Mediterranean", by Marilena Teri et al. (ref. egusphere-2024-701)**

*By François Dulac, LSCE, 06 December 2024*

**Recommendation**

Publication

**Reviewer Comments to Authors**

Synthesis:

I find that the revised manuscript has nicely solved the issues that the initial version raised to me. I consider that authors did a good job in clarifying and improving the manuscript, which I now find excellent in terms of significance, and both scientific and presentation quality. I definitely recommend publication.

I acknowledge that my minor comment on line 206 was a mistake, related to another manuscript. Just in case, I listed below a few editorial corrections that I noticed.

Editorial comments:

Line numbers refer to the track-change version.

- Line 185: a space should be shifted from after to before the citation of Schöberl et al.

- Line 299: spare space to be removed at end of line before the final dot.

- Line 305: rather replace "on" by "at".

- Line 325: "contents" (plural).

- Line 362: "to a higher" (see beginning of the sentence).

- Line 571: move Horvath et al. (2018) citation after Kim et al. (2011).

---

## Author Response (AR2)

Reply to the Editor concerning the minor revision for the manuscript: „Pollution affects Arabian and Saharan dust optical properties in the Eastern Mediterranean" by Teri et al.

We would like to thank the two reviewers and the editor for their time to review this manuscript, and for their comments which helped us to improve the manuscript.

In the following, the comments raised by the editor and reviewers are marked in blue. Our answers are given in black and also include a description of changes made to the manuscript.

Comments from the Editor:
Before submitting your production files, please shorten the abstract to maximum 250 words, as stated in the Author's guidelines.

The abstract was shortened.

Please consider the Notification to the authors from review file validation when preparing the production files.

The notification to the authors was considered. Please see below for details.
* * *
Notification to the authors from review file validation:

Please ensure that the color schemes used in your maps and charts allow readers with color vision deficiencies to correctly interpret your findings. Please check your figures using the Coblis – Color Blindness Simulator (https://www.color-blindness.com/coblis-color-blindness-simulator/) and revise the color schemes accordingly with the next file upload request.

Please re-check whether Figures 7, 8, 11 can be interpreted incorrectly by users with color vision deficiencies.

Figure 8a: The palette for the SSAAE was changed. We used the option turbo of the Viridis package, specifically designed to make reading easier for people with color blindness.

Figure 8b: The colors were slightly modified, and different shapes were used for the different pollution types.

Figure 7: The colors do not seem hard to read for people with color blindness. Thus, we left the color, but we used two different shapes for the two mineral dust types.

Figures 11 and 12: The colors were modified.
* * *
- Line 185: a space should be shifted from after to before the citation of Schöberl et al.
Done

- Line 299: spare space to be removed at end of line before the final dot.
The extra space seems to be present only in the track-change version.

- Line 305: rather replace "on" by "at".
Done

- Line 325: "contents" (plural).
Done

- Line 362: "to a higher" (see beginning of the sentence).
Done

- Line 571: move Horvath et al. (2018) citation after Kim et al. (2011).
Done
* * *
Additionally, we have also adapteed the references to the figures in the supplement to the new figure numbers.